# Activation of the myosin motors in fast-twitch muscle of the mouse is controlled by mechano-sensing in the myosin filaments

Cameron Hill[1] , Elisabetta Brunello[1] , Luca Fusi[1,2] , Jesús Garcia Ovejero[1] and Malcolm Irving[1]

[1]*Randall Centre for Cell & Molecular Biophysics, King's College London, London, UK*
[2]*Centre for Human & Applied Physiological Sciences, King's College London, London, UK*

Handling Editors: Michael Hogan & Wolfgang Linke

The peer review history is available in the Supporting information section of this article (https://doi.org/10.1113/JP283048#support-information-section).

The Journal of Physiology

**Abstract** Myosin motors in resting muscle are inactivated by folding against the backbone of the myosin filament in an ordered helical array and must be released from that conformation to engage in force generation. Time-resolved X-ray diffraction from single fibres of amphibian muscle showed that myosin filament activation could be inhibited by imposing unloaded shortening at the start of stimulation, suggesting that filaments were activated by mechanical stress. Here we improved the signal-to-noise ratio of that approach using whole extensor digitorum longus muscles of the mouse contracting tetanically at 28°C. Changes in X-ray signals associated with myosin filament activation, including the decrease in the first-order myosin layer line associated with the helical motor array, increase in the spacing of a myosin-based reflection associated with packing of myosin tails in the filament backbone, and increase in the ratio of the 1,1 and 1,0 equatorial reflections associated with movement of motors away from the backbone, were delayed by imposing 10-ms unloaded shortening

**Cameron Hill** obtained his Bachelor's degree in Sport and Exercise Sciences from Coventry University followed by a PhD in the same institution, performing skeletal muscle contractility measures on old and obese mice. After receiving his PhD in 2018 and following a research technician position at the Royal Veterinary College, he joined Prof. Malcolm Irving at King's College London as a postdoctoral researcher in 2019, using X-ray diffraction at synchrotrons to elucidate the regulatory mechanisms of muscle contraction in the thick filament of skeletal muscle. He aims to use X-ray diffraction to better understand the muscle ageing response in muscle.

at the start of stimulation. These results show that myosin filaments are predominantly activated by filament stress, as in amphibian muscle. However, a small component of filament activation at zero load was detected, implying an independent mechanism of partial filament activation. X-ray interference measurements indicated a switch-like change in myosin motor conformation at the start of force development, accompanied by transient disordering of motors in the regions of the myosin filament near its midpoint, suggesting that filament zonal dynamics also play a role in its activation.

(Received 16 March 2022; accepted after revision 27 July 2022; first published online 1 August 2022)

**Corresponding author** C. Hill: Randall Centre for Cell & Molecular Biophysics, King's College London, London, SE1 1UL, UK. Email: cameron.hill@kcl.ac.uk

**Abstract figure legend** Mechano-sensing in the thick filament of mammalian skeletal muscle. Time-resolved X-ray diffraction allows changes in the structure of the myosin-containing thick filaments and actin-containing thin filaments during contraction to be followed in an intact working muscle. In resting muscle (cyan), thin filaments are OFF because no calcium is bound to troponin (hexagons), and myosin motors are inhibited by folding back against the thick filament backbone in a helical array (grey). During contraction at high load (purple), most myosin motors leave the helical array, and some (green) bind to the thin filament to generate force, whilst others (yellow) become disordered. When a period of rapid shortening is imposed at the start of electrical stimulation to keep the load on the muscle very low (low-load; continuous black force trace), nearly all motors remain in the helical folded state (grey) even though the thin filament is ON and two calcium ions (white) are bound to each troponin (hexagons). These results show that activation of myosin motors in mammalian skeletal muscle is controlled by mechanical stress in the thick filament.

## Key points

- Activation of myosin filaments in extensor digitorum longus muscles of the mouse is delayed by imposing rapid shortening from the start of stimulation.
- Stress is the major mechanism of myosin filament activation in these muscles, but there is a small component of filament activation during electrical stimulation at zero stress.
- Myosin motors switch rapidly from the folded inhibited conformation to the actin-attached force-generating conformation early in force development.

## Introduction

Contraction of skeletal muscles is driven by the interaction between the thick (myosin-containing) and thin (actin-containing) filaments, a process that is triggered by an action potential in the surface membrane of the muscle cell, leading to the release of calcium ions from the sarcoplasmic reticulum and a transient increase in intracellular $[Ca^{2+}]$. The released calcium ions bind to troponin in the thin filament, resulting in structural changes in troponin and tropomyosin that uncover the myosin-binding sites on actin, permitting the interaction between myosin, actin and ATP that drives contraction (Huxley, 1973; Parry & Squire, 1973). Because the $[Ca^{2+}]_i$ transient is large and fast, it is expected to quickly saturate the binding sites on troponin (Baylor & Hollingworth, 2003). Therefore it seems likely that the calcium transient acts as a start signal for contraction but may not control the strength, speed or metabolic cost of contraction (Hill et al., 2021; Irving, 2017).

Recently it has become clear that those fundamental aspects of contraction are likely to be controlled by structural changes in the *thick* filament (Hill et al., 2021;

Irving, 2017; Linari et al., 2015). In resting skeletal muscle, the motor domains of myosin are folded back against the thick filament in quasi-helical tracks, so that they are not available for actin-binding (Craig & Padrón, 2022; Irving, 2017; Woodhead et al., 2005). This 'OFF' conformation of the myosin motors has been associated with a population of myosin motors that hydrolyse ATP extremely slowly, a 'super-relaxed state' of myosin that may have evolved to minimise the metabolic cost of resting muscle (Stewart et al., 2010).

The central unanswered question raised by this dual-filament paradigm of muscle regulation is then: what causes the myosin motors to leave the helical folded OFF state on the thick filament surface when the thin filament is activated by calcium binding? One possible answer was suggested by X-ray diffraction experiments on isolated fibres from frog skeletal muscle (Linari et al., 2015), which showed that the helical folded motor conformation characteristic of resting muscle could be retained after calcium activation of the thin filament if the load on the muscle was very low. In those conditions, the muscle shortens rapidly, but no external work is done, and stiffness measurements suggest that such shortening

may require only 1% of the myosin motors in the filament (Fusi et al., 2017).

It is possible then that a small fraction of the motors in each filament are not in the folded helical conformation in resting muscle, and that these 'constitutively ON' (Linari et al., 2015) or 'sentinel' motors (Craig & Padrón, 2022) could sense the activation of the thin filament. Moreover, if shortening of the muscle is prevented, resulting in an increased load on the muscle according to the well-known force–velocity relationship, many more motors leave the folded helical state, suggesting that thick filaments may be directly activated by mechanical stress (Linari et al., 2015). This thick filament mechano-sensing mechanism, which is independent of the intracellular calcium concentration in skeletal muscle (Fusi et al., 2016), would act to match the fraction of active myosin motors to the load, and may indeed provide a molecular explanation for the force–velocity relationship and the coupling between mechanical work and its metabolic cost (Linari et al., 2015).

Here, exploiting recent advances in synchrotron-based X-ray beamlines and detectors, we apply the protocol used by Linari et al. (2015) to determine the mechanism of activation of the thick filaments in fast-twitch mammalian skeletal muscles contracting in near-physiological conditions. We used isolated EDL muscles from the mouse, which diffract more strongly than the single fibres from frog muscle used in the previous studies. Despite the faster time course of activation of the mouse EDL muscles at near-physiological temperature, X-ray data with high signal-to-noise and spatial resolution could be collected with 5-ms time resolution (Hill et al., 2021). This allowed us to impose a 10-ms period of unloaded shortening at the start of electrical stimulation and follow the time course of the structural changes in the thick filaments during both the unloaded shortening and the subsequent period of fixed-end force development. The results provide strong evidence that the mechano-sensing mechanism operates in mammalian muscles in near-physiological conditions, but also revealed some smaller changes in thick filament structure during calcium activation at zero load and transient changes in the activation of different zones along the filament.

## Methods

### Animals

Male mice (strain C57BL/6J) aged 4−6 weeks were housed at MRC Harwell in groups of 4−5 in 12:12 h light–dark cycles at 50% relative humidity, with *ad libitum* access to water and a standard lab diet. All animals were housed and maintained in accordance with the ARRIVE 2.0 guidelines (Percie du Sert et al., 2020).

### Muscle preparation

Animals were sacrificed by cervical dislocation, followed by a confirmation method of permanent cessation of circulation, in compliance with the UK Home Office Animals (Scientific Procedures) Act 1986, Schedule 1. After sacrifice, whole extensor digitorum longus (EDL) muscles were carefully dissected from the hindlimb under a stereomicroscope in a trough continuously perfused with physiological solution (composition in mM: NaCl 118; KCl 4.96; $MgSO_4$ 1.18; $NaHCO_3$ 25; $KH_2PO_4$ 1.17; glucose 11.1; $CaCl_2$ 2.52) with pH ∼7.4 at room temperature after equilibration with carbogen (95% $O_2$, 5% $CO_2$). Metal hooks were tied with suture silk at the proximal and distal tendons of the muscle to allow attachment to the experimental set-up. The muscle was mounted in a custom 3D-printed plastic chamber between a fixed hook and the lever of a dual-mode force/length transducer (300C-LR, Aurora Scientific, Aurora, Canada), continuously perfused with physiological solution equilibrated with carbogen at 28°C.

Electrical stimuli were provided by a high-power biphasic stimulator (701C, Aurora Scientific) via parallel platinum electrodes attached to two mylar windows positioned as close as possible to the muscle to mini-mise the X-ray path in the solution. The stimulus voltage was set to 1.5 times that required to elicit the maximum force response. Muscle length was set to $L_0$, defined as that producing maximum force in response to a 100-ms train of stimuli at 130 Hz repeated at 5 minute intervals. $L_0$ was 12.2 ± 0.4 mm (mean ± SD; $n = 10$). Muscle cross-sectional area was estimated as $(2 \times (W_{MW}))/(\rho \times L_0)$, where $\rho = 1.06$ g $cm^{-3}$ is the density of the muscle and $W_{MW}$ is the muscle wet weight (Méndez & Keys, 1960). $W_{MW}$ was 10.0 ± 0.8 mg, giving a cross-sectional area of 1.55 ± 0.12 $mm^2$. Plateau force in the tetani that were fixed-end throughout was 272.6 ± 36.1 kPa (mean ± SD; $n = 5$) and that in tetani in which low-load shortening was imposed for 10 ms at the start of stimulation was 254.5 ± 14.2 kPa ($n = 5$); these values are not significantly different ($P = 0.327$, Student's *t*-test). Some X-ray results from the set of muscles used for the fixed-end protocol have been reported previously (Hill et al., 2021).

### X-ray data collection

The trough was sealed to prevent solution leakage and the muscle was mounted vertically at beamline I22 of the Diamond Light Source (Didcot, UK), to take advantage of the smaller vertical beam focus to optimise spatial resolution along the meridional axis (Bordas et al., 1995; Huxley et al., 2006; Linari et al., 2000). The mono-chromatic X-ray beam provided $6 \times 10^{12}$ photons $s^{-1}$ at 0.1 nm wavelength with full-width at half-maximum

about 300 $\mu$m horizontally and 100 $\mu$m vertically. X-ray diffraction patterns were recorded using a Pilatus P3-2M detector (Dectris Inc., Baden, Switzerland), of active area 253.7 × 288.8 mm, with 1475 × 1679 pixels, each 172 × 172 $\mu$m, organised in 3 × 8 modules (horizontal × vertical) with small gaps between modules (Fig. 1*A*). The sample-to-detector distance was set to 8.26 metres to optimise the position of the X-ray reflections of interest within the active area of the modules.

For muscle alignment in the X-ray beam, the beam was attenuated using a 0.1 mm molybdenum attenuator with transmission 0.0056, and regions of each muscle that produced relatively strong muscle-related and weak tendon-related X-ray reflections were identified. Rapid assessment of two-dimensional X-ray patterns was provided by the Data Analysis WorkbeNch software (DAWN; Basham et al., 2015).

Following muscle alignment, the attenuator was removed, and time-resolved two-dimensional patterns were collected in tetani that were fixed-end throughout ($n = 5$ muscles) or with a period of low-load shortening at the start of stimulation ($n = 5$ muscles). In the fixed-end protocol, the muscle remained at $L_0$ before electrical stimulation and was stimulated at $L_0$ for 100 ms. In the low-load shortening protocol, muscles were passively stretched by 5% of $L_0$ in 50 ms and held at that length for 300 ms before electrical stimulation, then allowed to shorten to $L_0$ in 10 ms from the first stimulus ($t = 0$), with

stimulation continuing for 70 ms. To minimise radiation damage, X-ray exposure was reduced to a minimum using a shutter and the muscle was moved vertically and/or horizontally between X-ray exposures. Data were acquired with 5-ms time resolution (3 ms acquisition and 2 ms readout time) for 340 ms (68 frames) in both protocols, with the first 90 ms (18 frames) of each X-ray exposure being required for shutter opening. Four resting frames were acquired in the 20 ms before the start of electrical stimulation. Signal-to-noise ratio was increased by signal-averaging 7–16 contractions per muscle in the low-load shortening protocol and 6–12 contractions in the fixed-end protocol, with peak force decreasing by 5.0 ± 2.1% in the low-load shortening protocol and 12.8 ± 0.5% in the fixed-end protocol between the first and last tetanus of the series.

Force, stimulus, muscle length and X-ray acquisition timing were sampled and analysed using custom-made software written in LabVIEW (National Instruments, Austin, TX, USA).

### X-ray data analysis

X-ray diffraction patterns were analysed using DAWN (Basham et al., 2015), SAXS package (P. Boesecke, ESRF, Grenoble, France), Fit2D (A. Hammersley, ESRF, Grenoble, France) and Igor Pro 8 (WaveMetrics, Inc., Lake Oswego, OR, USA). X-ray diffraction patterns containing

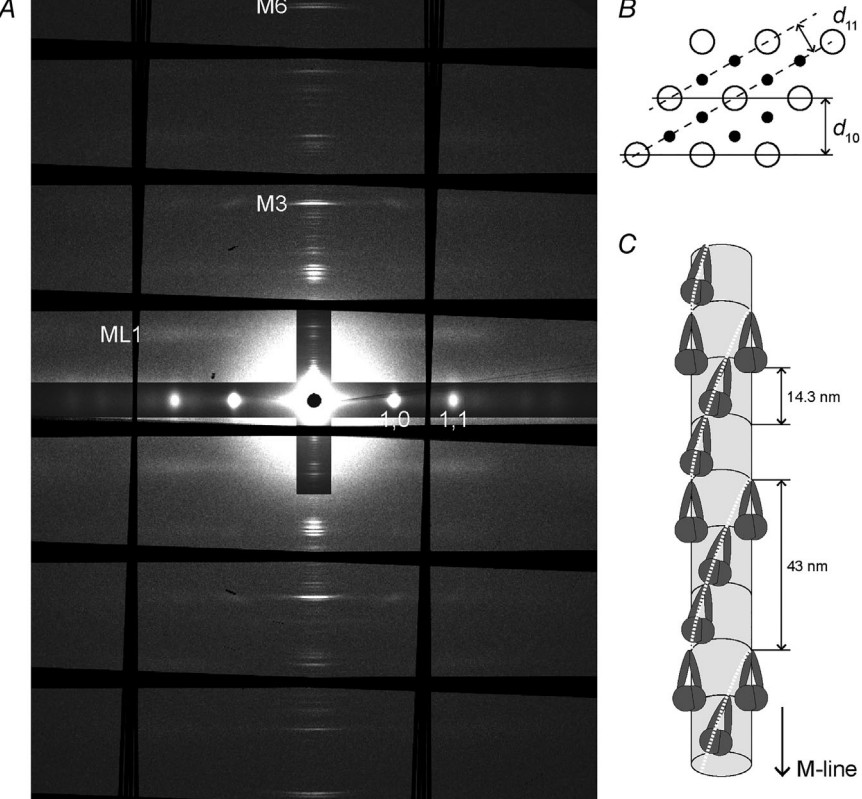

**Figure 1. X-ray diffraction pattern from resting EDL muscle**
*A*, small-angle X-ray diffraction pattern from resting mouse EDL muscles at 28°C, showing the meridional myosin-based reflections (M3 and M6), the first-order myosin layer line (ML1) and the 1,0 and 1,1 equatorial reflections. Data added from five muscles with total exposure time of 860 ms. The diffraction patterns were corrected for tilt using the 1,0 reflections but are unmirrored. The black regions arise from gaps between the detector tiles, with fanning introduced by the tilt correction. *B*, equatorial cross-section of the hexagonal lattice of thick filaments (open circles) and thin filaments (filled circles) showing the lattice planes associated with the 1,0 and 1,1 equatorial reflections. *C*, helical arrangement of myosin motor dimers (dark grey) in resting muscle with helical periodicity *ca* 43 nm, associated with the ML1 reflection in *A*. Each helical repeat contains three layers of dimers with axial periodicity 14.3 nm (associated with the M3 reflection), each folded against the thick filament backbone (light grey cylinder).

collagen-based reflections, indicating the presence of tendon in the X-ray beam, were excluded from further analysis. The series of 2D patterns from each contraction was corrected for camera background, added for each muscle, and centred and aligned using the equatorial 1,0 reflections (Fig. 1*A*).

For the analysis of the meridional reflections, aligned 2D patterns were mirrored horizontally and vertically. The distribution of diffracted intensity along the meridional axis of the X-ray pattern (parallel to the muscle axis) was calculated by integrating from 0.01522 nm$^{-1}$ on either side of the meridian. Background intensity distributions were fitted using a convex hull algorithm and subtracted; the small residual background was removed using the intensity distribution from a nearby region of the pattern containing no reflections. Integrated intensities were obtained from the following axial regions: M3, 0.066−0.072 nm$^{-1}$; M6, 0.135−0.142 nm$^{-1}$. The cross-meridional width of the M3 and M6 reflections was determined from the radial intensity distribution in the axial regions defined above using a single Gaussian centred on the meridian. The interference components of the M3 and M6 reflections were characterised by fitting multiple Gaussian peaks with the same axial width to the meridional intensity distribution. The total intensity of the meridional reflections was calculated as the sum of the intensity of the component peaks and multiplied by the cross-meridional width to correct for lateral misalignment between filaments during contraction (Huxley et al., 1982); the spacing was calculated as the weighted average of that of the component peaks. In some time frames an additional reflection, called the 'star' peak (Caremani et al., 2021), was observed on the low angle side of the M3 reflection. This peak is not considered to be a component of the M3 reflection and was not included in the present determination of the relative intensities and spacings of the component peaks of that reflection.

For the analysis of the layer line and equatorial reflections, the aligned 2D patterns were mirrored only horizontally and vertically, respectively, to minimise errors in the intensity measurements arising from the presence, on one side of the 2D pattern, of the gaps between detector modules close to reflections of interest. The intensities of the first myosin and first actin layer lines (ML1 and AL1) were calculated by integrating the radial region between 0.0638 and 0.0826 nm$^{-1}$ from the meridional axis. Due to the partial overlap of ML1 and AL1, integrated intensities were obtained from the axial region 0.017–0.033 nm$^{-1}$ for each muscle and separated by global Gaussian deconvolution of the time series data under the simplifying assumption that their spacings, $S_{ML1}$ and $S_{AL1}$ respectively, and axial widths do not change during contraction.

The equatorial intensity distribution was determined by integrating from 0.0036 nm$^{-1}$ on either side of the equatorial axis (perpendicular to the muscle axis) and the intensities and spacings of the 1,0, Z-disk, 1,1 and 2,0 reflections were determined by fitting four Gaussian peaks in the region 0.02–0.065 nm$^{-1}$ with the following constraint:

$$d_{1,0} = d_{1,0}/(3)^{1/2}, \, d_{2,0} = d_{1,0}/2 \text{ and } d_Z = d_{1,0} \times \text{constant}.$$

1D intensity distributions for each muscle were normalised to the intensity of the 1,0 at rest to correct for differences in the diffracting mass in the X-ray beam (Reconditi et al., 2014). The normalised 1D intensity distributions in Figs 2–5*A* and *B* and Fig. 6 were then averaged from the five muscles in each protocol.

## Interpretation of the axial profile of the M3 reflection

Myosin filaments from skeletal muscle are centrosymmetric about their midpoint at the M-band of the sarcomere and contain two arrays of 49 myosin motors with an axial periodicity *d* of about 14.5 nm, corresponding to the spacing of the M3 X-ray reflection ($S_{M3}$), with the first layer (layer 1) starting about 80 nm (the half bare zone, HBZ) from the filament midpoint. The centrosymmetric structure of the myosin filament generates interference fringes that sample the X-ray reflection produced by a single array of motors to give finely spaced sub-peaks (Brunello et al., 2020; Linari et al., 2000), from which the length of each array of diffracting motors contributing to the M3 reflection and the distance between their centres – the interference distance *D* – can be determined. For that purpose, each layer of myosin motors in the myosin filament was represented as a point diffractor. This gives an excellent approximation to the profile of the M3 reflection calculated from the full axial mass profile of the motors (Reconditi, 2006).

In the point diffractor representation, HBZ represents the average centre of mass of the first layer of myosin motors from the filament midpoint, and the diffracted intensity distribution along the meridional axis can be calculated as (Reconditi, 2006):

$$I(R) = [\sin(N\pi Rd) / \sin(\pi Rd)]^2 \times [\cos(\pi RD)]^2$$

where *R* is the reciprocal space co-ordinate, *N* is the number of layers of motors in a contiguous array from the first medial layer $n_m$ to the last distal layer $n_d$ that contribute to the M3 reflection, while layers outside that region make zero contribution because their motors are isotropic, and *D* is the interference distance calculated as $D = 2 \times \text{HBZ} + (n_m + n_d - 2) \times d$.

For fitting to the experimental M3 profiles, the calculated $I(R)$ was convoluted with a Gaussian function with sigma ~190 $\mu$m representing the combined point-spread function of the X-ray beam and detector. The best fit for each muscle and time point was determined

**Table 1. Half-times ($t_{1/2}$) of mechanical and structural signals during force development in tetani that are fixed-end throughout or with a period of low-load shortening at the start of stimulation**

| | Fixed-end $t_{1/2}$ (ms) | Low-load shortening $t_{1/2}$ (ms) |
|---|---|---|
| $S_{M6}$ | 8.1 ± 1.5[a] | 14.5 ± 0.6* |
| $A_{ML1}$ | 8.6 ± 2.1* | 15.1 ± 1.3* |
| $M_{M3}$ | 10.4 ± 1.2* | 16.8 ± 0.4* |
| $L_{M3}$ | 10.8 ± 1.5* | 16.8 ± 0.3* |
| $I_{1,1}/I_{1,0}$ | 11.3 ± 1.6* | 17.5 ± 1.3* |
| $\Delta z$ | 11.6 ± 0.1* | 16.4 ± 0.03[a] |
| $S_{M3}$ | 12.4 ± 1.8* | 18.7 ± 0.7* |
| $A_{AL1}$ | 15.3 ± 2.6[a] | 23.3 ± 1.2 |
| Force | 17.0 ± 1.5 | 22.4 ± 0.8 |
| $A_{M3}$ | 20.8 ± 2.5[a] | 25.7 ± 3.5 |

$t_{1/2}$ with respect to first stimulus determined by sigmoidal fitting. Means ± SD from five muscles per protocol except for $A_{AL1}$ for low-load shortening where $n = 3$. *$P < 0.05$ when comparing the $t_{1/2}$ to the $t_{1/2}$ of force in each protocol using a paired *t* test. [a]$P < 0.05$ for the corresponding comparison using the non-parametric Wilcoxon signed-rank test. $L_{M3}$ and $M_{M3}$, the fractional intensity of the low-angle and mid-angle peaks of the M3 reflection respectively.

by a global search of HBZ, *d*, $n_m$, $n_d$ and an intensity scaling factor, *y*, by minimising $\chi^2$ calculated using the experimental standard deviations from the difference between the experimental mean and the model output for the more intense peaks at each time point, corresponding to a reciprocal space region ranging between ∼0.068 and ∼0.071 nm$^{-1}$. To emphasise that changes in the fitted value of HBZ in the protocols used here report axial motion of the centre of mass of all the myosin motors in each half filament, they are reported in Fig. 7 as $\Delta z$, the increase in HBZ from its resting value.

### Statistical analyses

Data presented are means ± SD throughout. All data in Tables 1 and 2 were analysed using IBM SPSS Statistics (v27) (IBM Corp., Armonk, NY, USA). Data were first checked for normality of distribution using the Shapiro–Wilk test and homogeneity of variance using Levene's test to determine assumptions for parametric tests. Paired Student's *t*-tests were used to determine the significance of differences in the half-times of the changes in force and X-ray parameters. To determine differences between rest and subsequent time points, the four frames for data at rest were averaged and paired *t*-tests were conducted within each protocol. To compare differences between protocols at each time point, independent *t*-tests were used. In the event data were not normally distributed, the Wilcoxon signed-rank test was used for paired samples and the Mann–Whitney *U*-test for independent samples. Significance was set at $P < 0.05$ for all analyses.

## Results

### Force and shortening in fixed-end and low-load shortening protocols

When EDL muscles of the mouse were stimulated repetitively at a fixed length $L_0$ at 28°C (Fig. 2*C*, grey), force started to rise after a latency of 1.2 ± 0.1 ms and reached half of its tetanic plateau value by 17.0 ± 1.5 ms after the first stimulus (Table 1). When muscles were stretched to 1.05 $L_0$ before stimulation and allowed to shorten to $L_0$ at a constant velocity during the first 10 ms after the first stimulus (Fig. 2*C*, black), force stayed close to zero during the shortening, but started to increase almost immediately after the end of shortening and reached its half-maximum value 22.4 ± 0.8 ms after the first stimulus (Table 1). Thus, although the start of the force rise was delayed by 8.8 ms by low-load shortening, the time to half-maximum force was delayed by only 5.4 ms.

### Equatorial X-ray reflections

X-ray diffraction patterns were recorded at 5 ms intervals during the fixed-end and low-load shortening protocols. In the four resting frames collected before the stimulus, the equatorial 1,0 reflection from the hexagonal lattice of thick and thin filaments (Fig. 1*B*) was characteristically stronger than the 1,1 (Fig. 2*A* and *B*, cyan), signalling the proximity of the myosin motors to the thick filaments (Haselgrove & Huxley, 1973). At the tetanus plateau, the intensity of the 1,0 reflection became weaker and the 1,1 stronger and broader (Fig. 2*A* and *B*, red), signalling movement of the motors towards the thin filaments. The ratio of the intensities of the 1,1 and 1,0 reflections, $I_{1,1}/I_{1,0}$, was 0.3–0.4 at rest and increased to 1.8–1.9 at the tetanus plateau (Fig. 2*F*, red; Table 2), with half-times of 11.3 ± 1.6 ms and 17.5 ± 1.3 ms in the fixed-end and low-load shortening protocols respectively, leading force development in each case (Table 1). Although no change in $I_{1,1}/I_{1,0}$ was detectable in the first frame after the stimulus, which was collected from 0 to 3 ms, $I_{1,1}/I_{1,0}$ had already increased by the next frame, collected from 5 to 8 ms (Fig. 2*F*, light green; Table 2). Moreover, at that time, $I_{1,1}/I_{1,0}$ was already significantly larger in the fixed-end (Fig. 2*F*, open light green; Table 2) than in the low-load shortening protocol (filled light green), indicating greater movement of motors towards the thin filaments. This difference was also apparent in the intensity of the 1,0 reflection in the same time frame ($I_{1,0}$, Fig. 2*E*, filled light green).

**Table 2. Changes in force and structural parameters at rest, 5–8 ms, 10–13 ms, 15–18 ms after the first stimulus in the fixed-end and low-load shortening protocols, and at peak force ($T_0$)**

| | Fixed-end | | | | | Low-load shortening | | | | |
|---|---|---|---|---|---|---|---|---|---|---|
| | Rest | 5–8 ms | 10–13 ms | 15–18 ms | $T_0$ | Rest | 5–8 ms | 10–13 ms | 15–18 ms | $T_0$ |
| $T/T_0$ | 0.01 ± 0.001 | 0.16 ± 0.02*† | 0.33 ± 0.05*† | 0.51 ± 0.05*† | 1* | 0.02 ± 0.003 | −0.003 ± 0.006 | 0.03 ± 0.01* | 0.22 ± 0.02* | 1* |
| $d_{1,0}$ | 35.32 ± 0.24† | 35.68 ± 0.38* | 35.84 ± 0.38* | 36.00 ± 0.39* | 35.75 ± 0.36* | 34.13 ± 0.75 | 35.16 ± 0.71* | 35.96 ± 0.73* | 36.10 ± 0.65* | 36.05 ± 0.59* |
| $I_{1,0}$ | 1 | 0.82 ± 0.08*† | 0.66 ± 0.16*† | 0.56 ± 0.18* | 0.48 ± 0.14* | 1 | 0.96 ± 0.03* | 0.84 ± 0.09* | 0.62 ± 0.14* | 0.36 ± 0.16* |
| $I_{1,1}/I_{1,0}$ | 0.41 ± 0.04 | 0.77 ± 0.09*† | 1.19 ± 0.27*† | 1.45 ± 0.32b* | 1.77 ± 0.31* | 0.29 ± 0.09 | 0.44 ± 0.11* | 0.63 ± 0.13* | 0.98 ± 0.13* | 1.92 ± 0.34* |
| $I_{ML1}$ | 1 | 0.61 ± 0.23*† | 0.23 ± 0.13*† | 0.12 ± 0.15* | 0.09 ± 0.05* | 1 | 0.91 ± 0.12 | 0.66 ± 0.11* | 0.27 ± 0.10* | 0.10 ± 0.08* |
| $A_{ML1}$ | 1 | 0.77 ± 0.15*† | 0.47 ± 0.12*† | 0.38 ± 0.15* | 0.34 ± 0.11* | 1 | 0.95 ± 0.07 | 0.81 ± 0.07* | 0.51 ± 0.10* | 0.20 ± 0.07* |
| $I_{AL1}$ | 1 | 1.02 ± 0.46 | 1.44 ± 0.94a | 1.90 ± 0.87a | 2.68 ± 1.11* | 1 | 0.85 ± 0.66 | 0.99 ± 0.46 | 1.05 ± 0.68 | 2.58 ± 1.64 |
| $A_{AL1}$ | 1 | 0.99 ± 0.24 | 1.16 ± 0.35 | 1.36 ± 0.29* | 1.61 ± 0.33* | 1 | 0.86 ± 0.37 | 0.97 ± 0.23 | 0.98 ± 0.35 | 1.52 ± 0.57 |
| $I_{M6}$ | 1 | 1.16 ± 0.18 | 1.01 ± 0.22 | 1.03 ± 0.46 | 1.09 ± 0.47 | 1 | 1.05 ± 0.33 | 1.04 ± 0.32 | 1.06 ± 0.84 | 1.53 ± 1.98a |
| $S_{M6}$ | 7.173 ± 0.005 | 7.222 ± 0.012*† | 7.253 ± 0.009*† | 7.268 ± 0.011*† | 7.283 ± 0.003* | 7.176 ± 0.003 | 7.188 ± 0.004* | 7.202 ± 0.005* | 7.244 ± 0.009* | 7.280 ± 0.006* |
| $I_{M3}$ | 1 | 0.75 ± 0.16* | 0.99 ± 0.04b | 1.54 ± 0.13*b | 3.18 ± 0.56*b | 1 | 0.80 ± 0.12* | 0.73 ± 0.15* | 1.08 ± 0.75 | 3.86 ± 2.67* |
| $A_{M3}$ | 1 | 0.86 ± 0.09* | 0.99 ± 0.02b | 1.24 ± 0.05*b | 1.78 ± 0.16*b | 1 | 0.89 ± 0.06* | 0.85 ± 0.09* | 0.99 ± 0.36 | 1.86 ± 0.70 |
| $L_{M3}$ | 0.04 ± 0.02† | 0.07 ± 0.03b | 0.38 ± 0.12*† | 0.55 ± 0.07*† | 0.54 ± 0.01* | 0.06 ± 0.02 | 0 ± 0* | 0.03 ± 0.03 | 0.26 ± 0.05* | 0.58 ± 0.03* |
| $M_{M3}$ | 0.83 ± 0.02b | 0.79 ± 0.01*† | 0.54 ± 0.08*† | 0.39 ± 0.06*† | 0.43 ± 0.01* | 0.81 ± 0.04 | 0.84 ± 0.02 | 0.82 ± 0.02 | 0.64 ± 0.07* | 0.41 ± 0.02* |
| $H_{M3}$ | 0.13 ± 0.01 | 0.14 ± 0.02 | 0.09 ± 0.04b | 0.05 ± 0.02*† | 0.04 ± 0.003*† | 0.13 ± 0.02 | 0.16 ± 0.02 | 0.15 ± 0.01 | 0.10 ± 0.04 | 0.04 ± 0.01a |
| $S_{M3}$ | 14.339 ± 0.008 | 14.347 ± 0.011† | 14.431 ± 0.034*† | 14.490 ± 0.021*† | 14.533 ± 0.003* | 14.341 ± 0.010 | 14.312 ± 0.007* | 14.316 ± 0.010* | 14.394 ± 0.014* | 14.534 ± 0.011* |
| $d$ | 14.343 ± 0.009 | 14.354 ± 0.009† | 14.443 ± 0.042*† | 14.515 ± 0.018*† | 14.552 ± 0.003* | 14.343 ± 0.013 | 14.317 ± 0.003* | 14.322 ± 0.006* | 14.400 ± 0.009* | 14.550 ± 0.007* |

Rest denotes the average of the four frames from −18.5 to −3.5 ms, $T_0$ that from 61.5 to 76.5 ms for the fixed-end protocol and 51.5–66.5 ms for the low-load shortening protocol. Means ± SD for $n = 5$. *$P < 0.05$ when comparing a given time point to rest for each protocol using a paired $t$-test. †$P < 0.05$ when comparing fixed-end and low-load shortening protocols at given time points using an independent samples $t$-test. a$P < 0.05$ when comparing a given time point to rest for each protocol using the non-parametric Wilcoxon signed-rank test. b$P < 0.05$ when comparing fixed-end and low-load shortening protocols at given time points using the non-parametric Mann–Whitney $U$-test. $L_{M3}$, $M_{M3}$ and $H_{M3}$, the fractional intensity of the low-angle, mid-angle and high-angle peaks of the M3 reflection, respectively.

The spacing of the hexagonal filament lattice ($d_{1,0}$) was smaller at rest in the low-load shortening protocol (Fig. 2D, filled cyan) than in the fixed-end tetanus (open cyan), as expected from the greater sarcomere length before stimulation in that protocol. After low-load shortening, $d_{1,0}$ values for the two protocols were almost equal, as expected if the muscles had reached the same sarcomere length.

## Layer line X-ray reflections

In resting muscle, myosin motors are folded back against their tails and form a helical array with a periodicity of *ca* 43 nm that gives rise to a strong off-axial or layer-line X-ray reflection called ML1 (Fig. 1A and C), and weaker higher orders of that reflection (Caremani et al., 2019; Huxley & Brown, 1967; Ma et al., 2020). ML1 (Fig. 3A and B; centred on the vertical dashed line) partially overlaps the actin-based layer line (AL1; centred on the continuous vertical line) associated with the *ca* 37 nm helical periodicity of the thin filament, but the ML1 and AL1 components can be separated by global Gaussian deconvolution of the time series data under the assumption that their periodicities $S_{ML1}$ and $S_{AL1}$ are invariant (Hill et al., 2021). This procedure gave $S_{ML1}$ values of $43.0 \pm 0.3$ and $42.8 \pm 0.1$ nm in the fixed-end and low-load shortening protocols, respectively. The corresponding values for $S_{AL1}$ were $36.6 \pm 0.3$ and

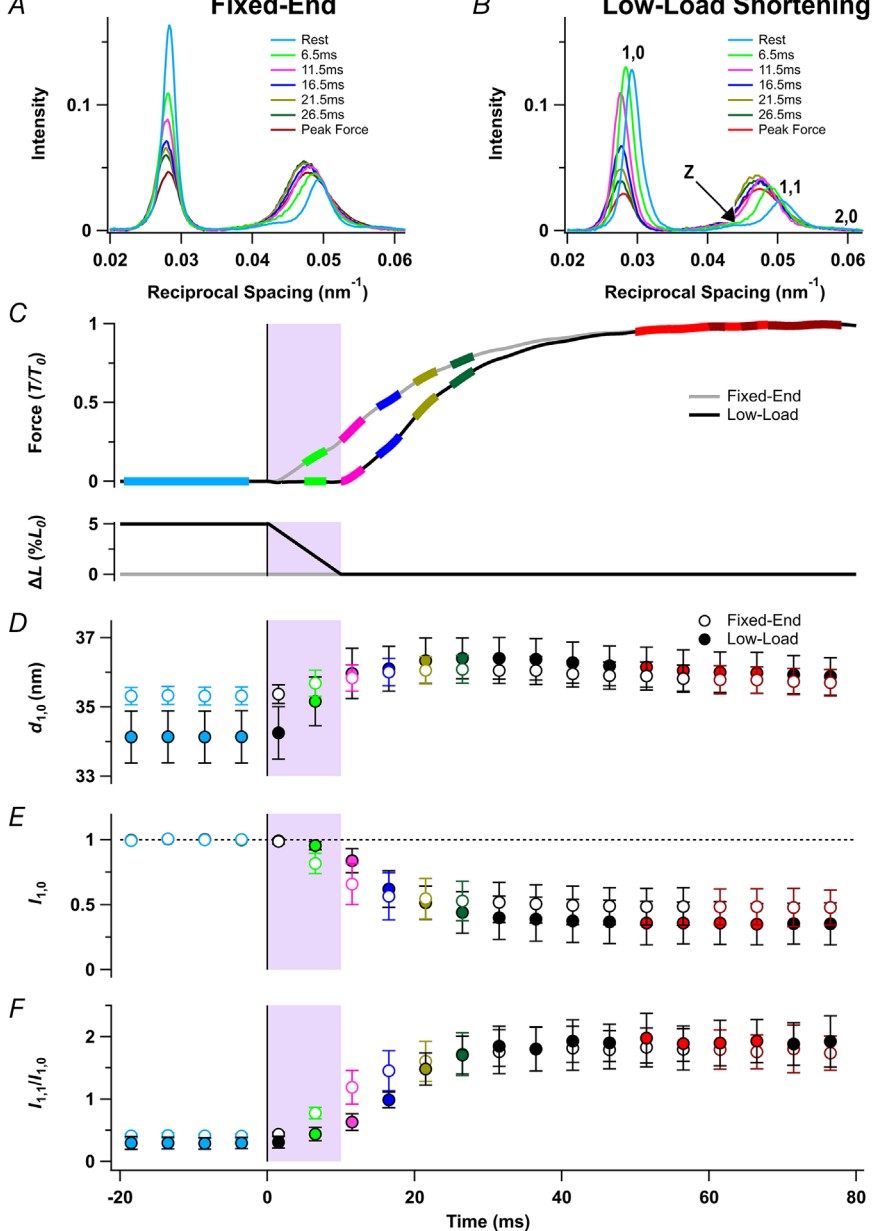

**Figure 2. Equatorial X-ray reflections**
*A* and *B*, distribution of intensity along the equator of the diffraction pattern, perpendicular to the muscle long axis, in tetani that are fixed-end throughout (*A*) or with a period of low-load shortening at the start of stimulation (*B*). Cyan, rest, average of four frames centred on −18.5, −13.5, −8.5 and −3.5 ms; light green, 6.5 ms; magenta, 11.5 ms; blue, 16.5 ms; gold, 21.5 ms; dark green, 26.5 ms; red, average of four frames centred on 51.5, 56.5, 61.5 and 66.5 ms, at the force plateau in the low-load shortening protocol; dark red, average of four frames centred on 61.5, 66.5, 71.5 and 76.5 ms, at the force plateau in the fixed-end protocol. *C–F*, time course of changes in relative force ($T/T_0$) and change in length as a percentage of $L_0$ (*C*) in the fixed-end protocol (grey) and the low-load shortening protocol (black), lattice spacing ($d_{1,0}$; *D*), intensity of the 1,0 reflection relative to rest ($I_{1,0}$; *E*) and equatorial intensity ratio ($I_{1,1}/I_{1,0}$; *F*). Open and filled symbols in *D–F* denote the fixed-end and low-load shortening protocols, respectively; colours denote the time periods used to calculate the profiles in *A* and *B*. The shaded areas denote the 10-ms period of low-load shortening following the first stimulus at *t* = 0 (vertical black line). Data presented as means ± SD from five muscles per protocol. Dashed horizontal black line in *E* denotes resting value. [Colour figure can be viewed at wileyonlinelibrary.com]

$36.9 \pm 0.8$ nm. $S_{ML1}$ and $S_{AL1}$ were not significantly different between protocols ($P = 0.329$ and $P = 0.551$, respectively).

The intensity of the ML1 reflection ($I_{ML1}$) decreased rapidly during fixed-end activation (Fig. 3*D*, open circles), and its value at the tetanus plateau was only about 10% of that at rest. The best estimate of the fraction of motors in the folded helical conformation is given by the square root of $I_{ML1}$, or $A_{ML1}$ (Fig. 3*F*) which had a half-time of only $8.6 \pm 2.1$ ms in the fixed-end protocol (Table 1). $A_{ML1}$ during low-load shortening (Fig. 3*F*, filled light green) remained close to its resting value (filled cyan), and the overall half-time of its decrease during activation was $15.1 \pm 1.3$ ms, again faster than the rise

of force. $I_{AL1}$ increased by about a factor of three during activation (Fig. 3*E*, red), as myosin motors attach to thin filaments (Hill et al., 2021), and the half-time of $A_{AL1}$ in the fixed-end protocol was $15.3 \pm 2.6$ ms, not significantly different from that of force (Table 1). $A_{AL1}$ did not change significantly during the low-load shortening (Table 2) and the overall half-time of its increase in the low-load shortening protocol was $23.3 \pm 1.2$ ms, again not significantly different from that of force (Table 1).

### The meridional M6 X-ray reflection

The meridional M6 reflection (Fig. 1*A*) is associated with the ~7.2 nm axial periodicity of the thick filament

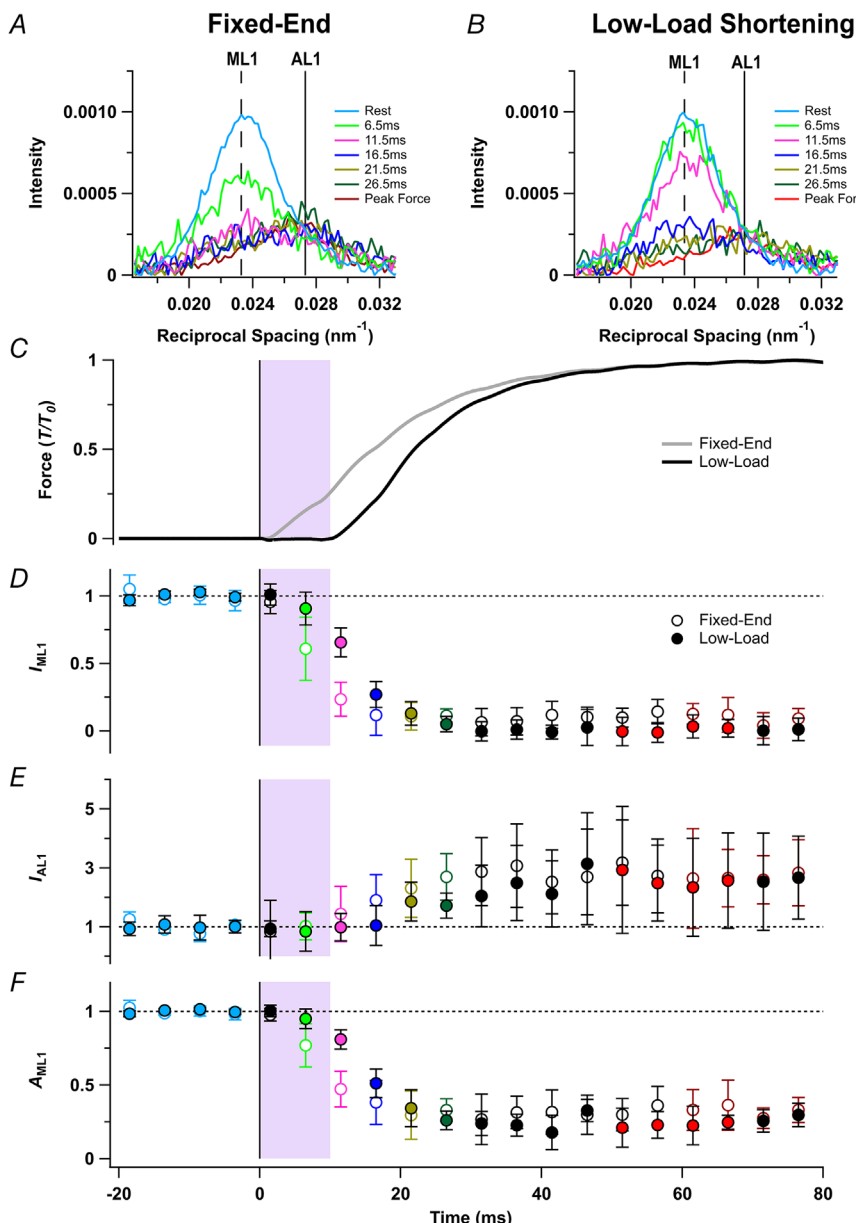

**Figure 3. Myosin and actin X-ray layer line reflections**

*A* and *B*, axial distribution of diffracted intensity in the region of the mixed myosin/actin first layer line in tetani that are fixed-end throughout (*A*) or with a period of low-load shortening at the start of stimulation (*B*). Cyan, rest; light green, 6.5ms; magenta, 11.5 ms; blue, 16.5 ms; gold, 21.5 ms; dark green, 26.5 ms; red, force plateau in low-load shortening protocol; dark red, force plateau in the fixed-end protocol. Vertical dashed and continuous lines denote global best-fit axial reciprocal spacings of the myosin- (ML1) and actin-based (AL1) layer lines, respectively. *C*–*F*, time course of changes in relative force ($T/T_0$; *C*) for the fixed-end (grey) and low-load shortening (black) protocols, intensity of the first myosin layer line ($I_{ML1}$; *D*), intensity of the first order actin layer line ($I_{AL1}$; *E*), and the amplitude of the first order actin layer line ($A_{ML1}$; *F*) normalised to respective mean resting values. Open and filled symbols in *D*–*F* denote the fixed-end tetanus and low-load shortening protocols respectively; colours denote the time periods used to calculate the profiles in *A* and *B*. The shaded areas denote the 10-ms period of low-load shortening following the first stimulus at $t = 0$ (vertical black line). Data presented as means $\pm$ SD from five muscles per protocol. Dashed horizontal black lines denote resting values. [Colour figure can be viewed at wileyonlinelibrary.com]

backbone (Huxley et al., 2006; Reconditi et al., 2004). Two sub-peaks can be identified in most conditions, and these are likely to be due to X-ray interference between the two halves of each thick filament (Fig. 4*A* and *B*). The lower angle sub-peak (LA) is more intense than the higher angle (HA) sub-peak. The intensity of the M6 reflection ($I_{M6}$; Fig. 4*C*) did not change significantly during activation in either the fixed-end (open circles) or the low-load shortening (filled circles) protocol, but its spacing ($S_{M6}$; Fig. 4*D*) increased on activation by *ca* 1.5% in both protocols. The half-time of the increase in $S_{M6}$ in the fixed-end protocol was $8.1 \pm 1.5$ ms (Table 1). The increase was slower during low-load shortening (half-time $14.5 \pm 0.6$ ms; Table 1), but there was a significant increase

in $S_{M6}$ during low-load shortening (Fig. 4*D*, filled light green; Table 2), albeit significantly less than at the same time in the fixed-end protocol (open light green; Table 2).

The interference sub-peaks of the M6 reflection showed systematic changes during early activation (Fig. 4*E* and *F*). The fractional intensity of the HA peak is similar at rest (Fig. 4*E*, cyan triangles) and the tetanus plateau (red triangles) in both protocols, but transiently decreases around 10 ms after the first stimulus, and this decrease is larger in the low-load shortening protocol, notably in the 10−13 ms frame immediately after low-load shortening (filled magenta triangle). The spacings of the LA and HA peaks increased on activation roughly in parallel with $S_{M6}$ (Fig. 4*F*).

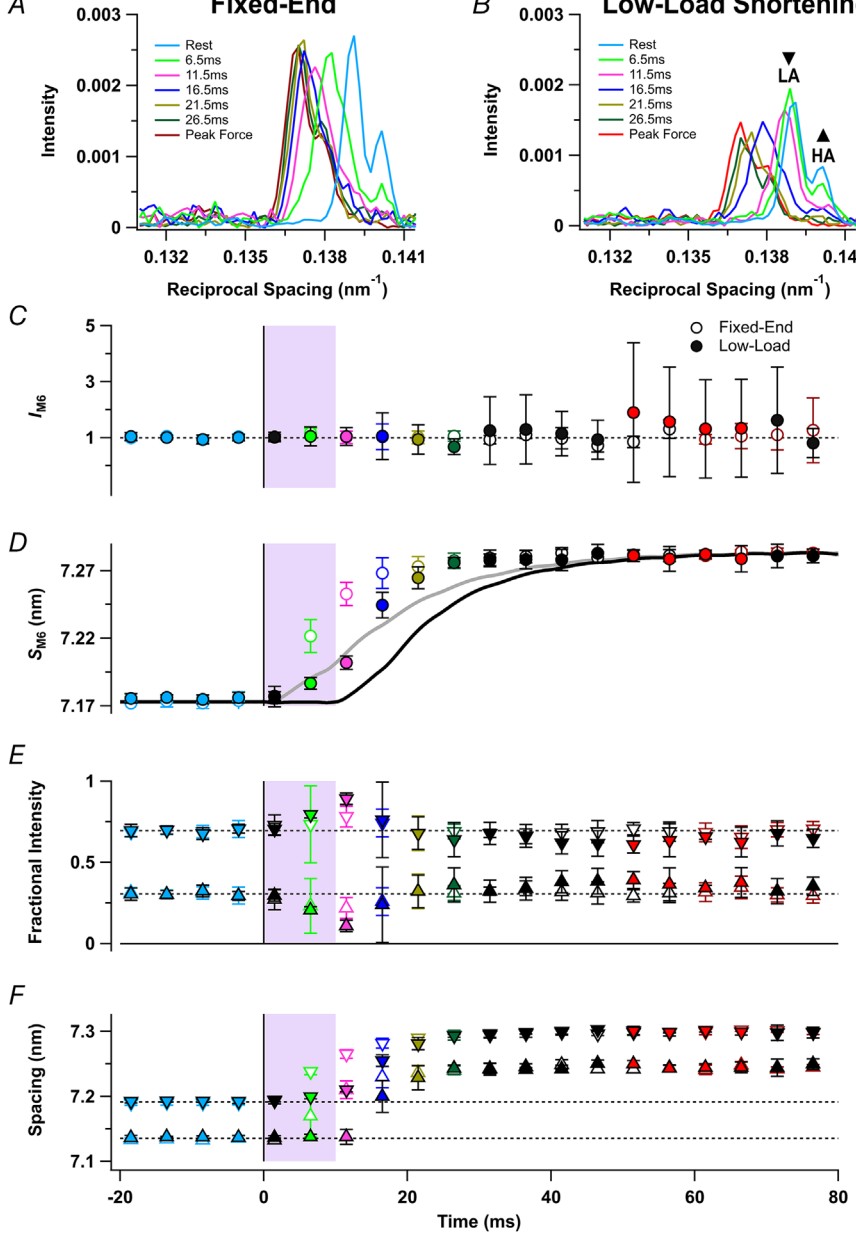

**Figure 4. The M6 X-ray reflection**
*A* and *B*, axial distribution of diffracted intensity in the region of the M6 reflection in tetani that are fixed-end throughout (*A*) or with a period of low-load shortening at the start of stimulation (*B*). Cyan, rest; light green, 6.5 ms; magenta, 11.5 ms; blue, 16.5 ms; gold, 21.5 ms; dark green, 26.5 ms; red, force plateau in the low-load shortening protocol; dark red, force plateau in the fixed-end protocol. *C* and *D*, time-course of changes in the intensity of the M6 reflection ($I_{M6}$; *C*) normalised to its resting value, and the spacing of the M6 reflection ($S_{M6}$) superimposed on force for the fixed-end (grey) and low-load shortening (black) protocols (*D*). *E* and *F*, fractional intensity (*E*) and spacing (*F*) of the lower- (inverted triangles; LA) and higher-angle (triangles; HA) sub-peaks. Open and filled symbols in *C*–*F* denote fixed-end tetanus and low-load shortening protocols respectively; colours denote the time periods in *A* and *B*. The shaded areas denote the 10-ms period of low-load shortening following the first stimulus at $t = 0$ (vertical black line). Data presented as means ± SD from five muscles per protocol. Dashed horizontal black lines denote resting values. Continuous horizontal black line in *E* denotes zero fractional intensity. [Colour figure can be viewed at wileyonlinelibrary.com]

### The meridional M3 X-ray reflection

The meridional M3 reflection is associated with the axial periodicity of the myosin motors along the thick filaments (Fig. 1*A* and *C*), and changes in the intensity, spacing and interference fine structure of this reflection can be used to quantify changes in the conformation of the motors (Irving et al., 2000; Piazzesi et al., 2007; Reconditi et al., 2011). Three sub-peaks of the M3 reflection can be identified in resting muscle (Figs 5 and 6*A* and *B*, cyan), designated low- (LA), mid- (MA) and high-angle (HA), but the MA peak is the most intense (Caremani et al., 2019; Linari et al., 2000; Reconditi et al., 2011). At the tetanus plateau (Fig. 5*A* and *B*, red), the reflection is dominated by

the LA and MA sub-peaks, and the HA sub-peak becomes weak.

The intensity of the M3 reflection ($I_{M3}$; Fig. 5*C*) decreases transiently on activation, before increasing to about three times its resting value at the tetanus plateau (Fig. 5*C*, red) (Caremani et al., 2019; Reconditi et al., 2011). The transient decrease in $I_{M3}$ signals motors leaving the folded helical state of resting muscle, and the subsequent increase signals their binding to actin with their long axes roughly perpendicular to the filaments (Hill et al., 2021). Low-load shortening delayed the subsequent increase during fixed-end force development.

The spacing of the M3 reflection ($S_{M3}$) was about 1.4% larger at the tetanus plateau (Fig. 5*D*, red) than at

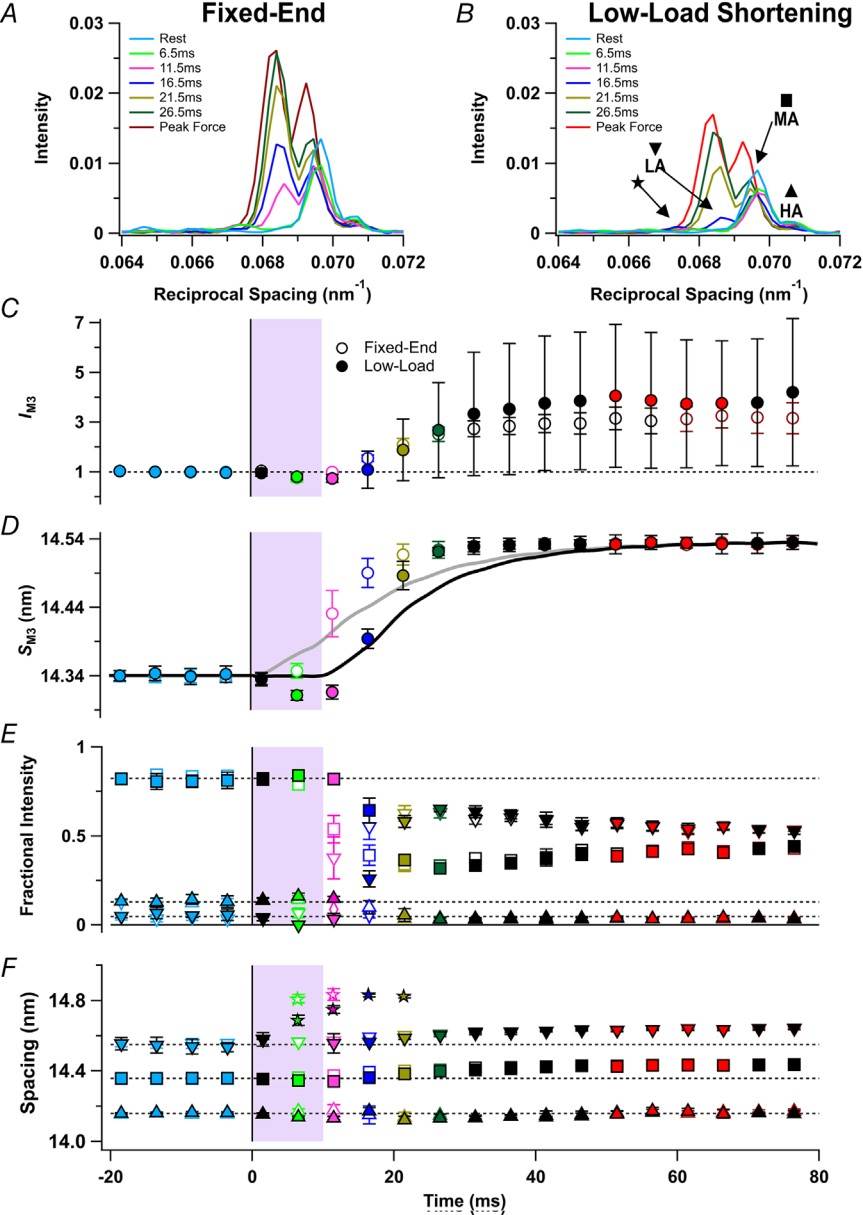

**Figure 5. The M3 X-ray reflection**
*A* and *B*, axial distribution of diffracted intensity in the region of the M3 reflection in tetani that are fixed-end throughout (*A*) or with a period of low-load shortening at the start of stimulation (*B*). Cyan, rest; light green, 6.5 ms; magenta, 11.5 ms; blue, 16.5 ms; gold, 21.5 ms; dark green, 26.5 ms; red, force plateau in the low-load shortening protocol; dark red, force plateau in the fixed-end protocol. *C* and *D*, time course of changes in the intensity of the M3 reflection ($I_{M3}$; *C*) normalised to its mean resting value and the spacing of the M3 reflection ($S_{M3}$) superimposed on force for the fixed-end (grey) and low-load shortening (black) protocols (*D*). *E* and *F*, fractional intensity (*E*) and spacing (*F*) of the 'star peak' (stars), and low-angle (inverted triangles; LA), mid-angle (squares; MA) and high-angle (triangles; HA) sub-peaks of the M3 reflection, with symbols defined in *B*. Fractional intensities of the M3-sub-peaks and the star peak were calculated with respect to $I_{M3}$. Open and filled symbols in *C–F* denote fixed-end and low-load shortening protocols; colours denote the time periods in *A* and *B*. The shaded areas denote the 10-ms period of low-load shortening following the first stimulus at $t = 0$ (vertical black line). Data presented as means ± SD from five muscles per protocol. Dashed horizontal black lines denote resting values. Continuous horizontal black line in *E* denotes zero fractional intensity. [Colour figure can be viewed at wileyonlinelibrary.com]

rest (cyan), slightly smaller than the percentage increase in $S_{M6}$. The increase in $S_{M3}$, which had a half-time of $12.4 \pm 1.8$ ms in the fixed-end protocol (Table 1), was delayed until after the end of low-load shortening (Fig. 5D, filled circles), and the half-time of its subsequent increase was faster than the force rise in the same protocol (Table 1). $S_{M3}$ during low-load shortening is slightly, but reproducibly, smaller than at rest, by about 0.2% (Fig. 5D, filled light green; Table 2).

The fractional intensity of the HA sub-peak of the M3 reflection shows small changes during early activation

(Fig. 5E, triangles). The half-time for the decrease in the fractional intensity of the MA sub-peak (squares) was similar to that of the increase in the LA peak (inverted triangles) in the fixed-end protocol, but both were faster than force (Table 1). Low-load shortening delayed all these changes, but their sequence remained the same (Fig. 5E, Table 1). The spacings of the LA and MA sub-peaks (Fig. 5F, inverted triangles and squares) increased during activation in a similar manner for both protocols.

A new peak, with a spacing of *ca* 14.8 nm and labelled 'star' in Fig. 5B, appeared in the 5−8 ms frame in the

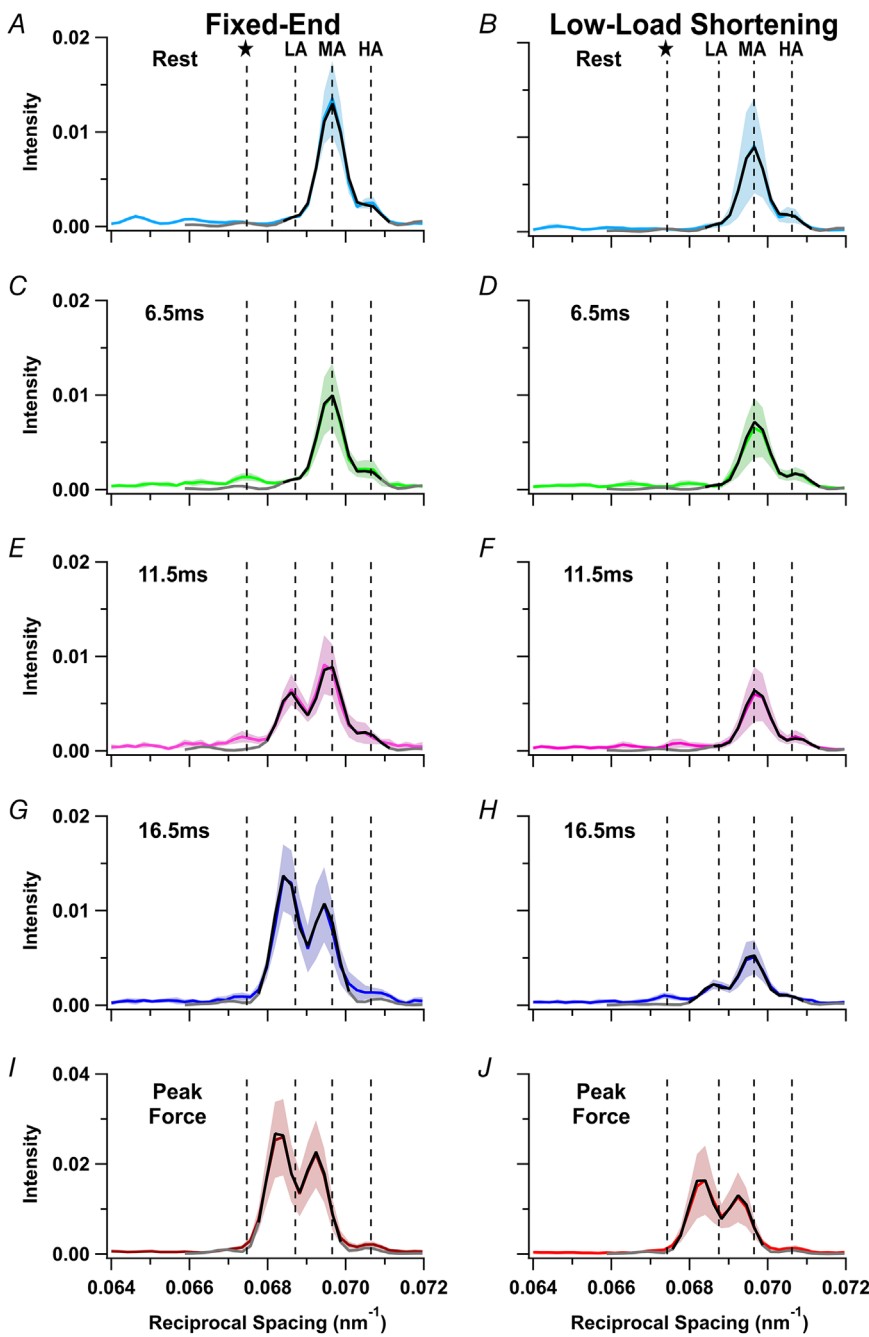

**Figure 6. Meridional X-ray intensity distributions in the region of the M3 reflection**

The experimental distributions are shown in colour with the shaded regions representing means ± SD; the best fits from the model are shown as continous lines for two regions of reciprocal space (black, narrow axial fit; grey, wide axial fit) for the fixed-end (*A*, *C*, *E*, *G* and *I*) and low-load shortening protocols (*B*, *D*, *F*, *H* and *J*). Rest (*A* and *B*, cyan); 6.5 ms (*C* and *D*, light green); 11.5 ms (*E* and *F*, magenta); 16.5 ms (*G* and *H*, blue); plateau force (*I* and *J*, red). Experimental data from five muscles per protocol except for panel *H* (four muscles per protocol). Vertical dashed lines denote the average position of the low-angle (LA), mid-angle (MA), and high-angle (HA) peaks, and the expected position of the 'star' (star) peak, at rest for each protocol. [Colour figure can be viewed at wileyonlinelibrary.com]

fixed-end tetanus (Fig. 5*F*, open light green star) (Hill et al., 2021) and lasted for one further frame (open magenta star). A peak with a spacing of *ca* 14.7 nm which we also associate with 'star' appeared in the 5−8 ms frame during low-load shortening (Fig. 5*F*, filled light green star), when no LA peak was detectable. This 'star' peak remained for the following three frames, in which its spacing was close to 14.8 nm (Fig. 5*F*, filled stars).

### Modelling the axial profiles of the M3 reflections

The relative spacing and intensities of the sub-peaks of the M3 reflection provide structural information regarding the axial conformation of the diffracting structures, the myosin motors, and their location within the thick filament (Brunello et al., 2020; Caremani et al., 2021; Hill et al., 2021; Ovejero et al., 2022). Each half filament

contains 49 layers of motors with axial periodicity $d$, starting from the edge of the half bare zone (a distance HBZ from the filament mid-point) and extending to the filament tip (Fig. 7*C*, inset). Only the motors from layer $n_m$ to layer $n_d$ are considered to be axially ordered, and therefore to contribute to the M3 reflection. The parameters HBZ, $d$, $n_m$ and $n_d$ were adjusted to give the best fit to the central region of the axial profile of the M3 reflection (Fig. 6, black) in order to avoid a possible contribution from the 'star' peak. The calculated profiles, extended to a wider region as the grey lines in Fig. 6, do not reproduce the 'star' peak (Hill et al., 2021), which seems to be associated with diffracting structures with a longer axial periodicity (Caremani et al., 2021).

In all cases, the best-fit value of $d$ was close to the experimental value of $S_{M3}$ (Fig. 7*B*; Table 2). The best-fit value of the parameter HBZ gives a precise measure

**Figure 7. Structural interpretation of changes in the fine structure of the M3 X-ray reflection**
Time course of changes in the best-fit model parameters from the fits in Figure 6 (black) for the average centre-of-mass of the myosin motors ($\Delta z$; *A*), the axial motor periodicity ($d$; *B*), and the most medial (squares; $n_m$) and distal (circles; $n_d$) diffracting layers in the half thick filament (*C*). The right-hand inset in *C* shows a schematic representation of the half filament, with black bars denoting layers of myosin motors with spacing $d$ numbered 1−49 starting from the edge of the half bare zone. Open and filled symbols in *A–C* denote the fixed-end and low-load shortening protocols, respectively; colours denote time points relative to data in Figure 6. Cyan, rest; light green, 6.5 ms; magenta, 11.5 ms; blue, 16.5 ms; gold, 21.5 ms; dark green, 26.5 ms; red, low-load shortening force plateau; dark red, fixed-end force plateau. Means ± SD from five muscles per protocol, except $n = 4$ for 16.5 ms of the low-load shortening protocol. The dashed horizontal black line in *B* denotes resting HBZ. Dashed lines in *C* denote boundaries between the P-zone (layers 1−12), MyBP-C-containing C-zone (12−30) and D-zone (30−49). [Colour figure can be viewed at wileyonlinelibrary.com]

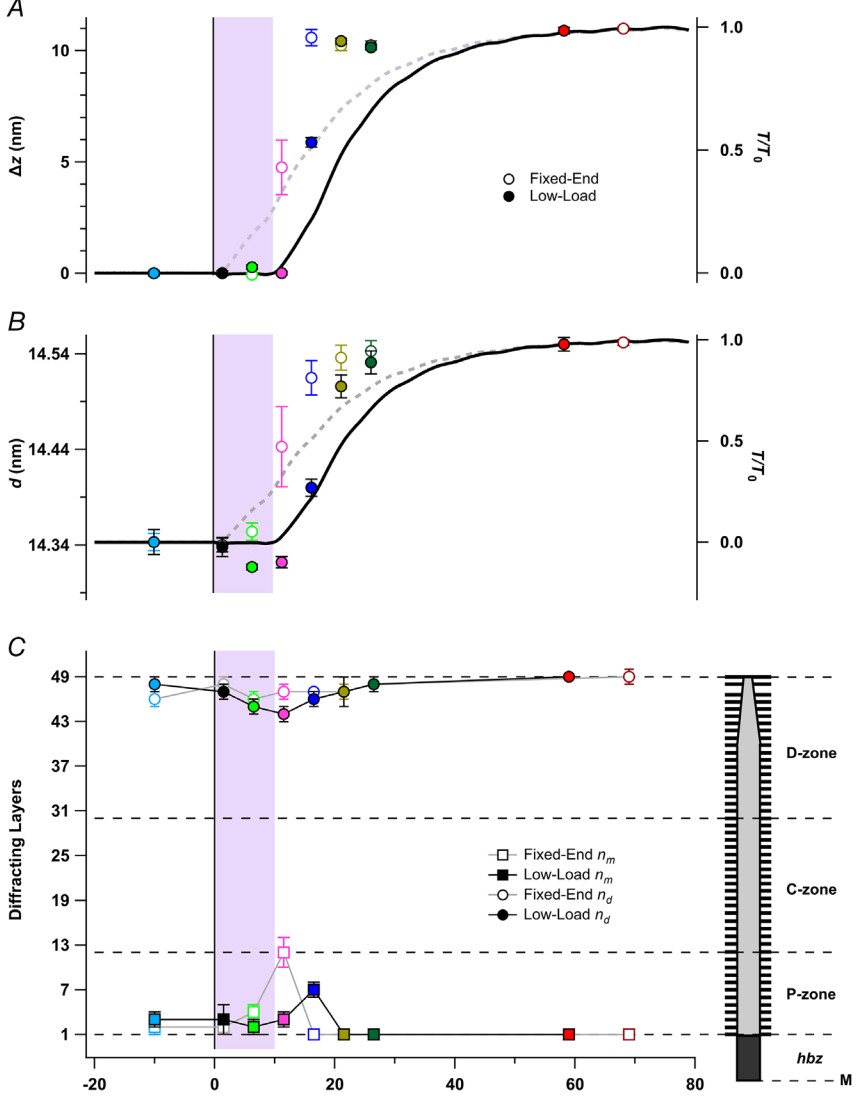

of the distance between the average centre of mass of the diffracting motors in the first myosin layer and the filament midpoint, and therefore of the motion ($\Delta z$) of the average centre of mass of all diffracting motors from their periodic attachment to the filament backbone (Fig. 7A). $\Delta z$ increases by about 11 nm during the development of tetanic force, signalling the large difference between the conformation of the folded motors and actin-attached isometric force-generating motors (Hill et al., 2021; Reconditi et al., 2011). However, $\Delta z$ did not increase during low-load shortening or in the first time frame after the end of shortening (Fig. 7A, filled green and magenta), but then switched almost to its tetanus plateau value in the next 10 ms (filled gold). $n_m$ and $n_d$ (Fig. 7C, squares and circles, respectively) indicate that nearly all 49 layers of motors in the half filament are axially ordered in resting muscle and at the tetanus plateau. $n_d$ decreased transiently during and immediately after unloaded shortening (filled circles), suggesting that a few layers of motors near the tip of the filament left the folded state (Hill et al., 2021). $n_m$ (squares) increased transiently in both the fixed-end and low-load shortening protocols, but the changes were delayed until after the shortening in the latter protocol.

## Discussion

The main aim of the experiments described above was to investigate the paradigm of thick filament mechano-sensing, in which activation of thick filaments and myosin motors depends on filament stress, in fast-twitch mammalian muscle. To that end, we imposed a period of unloaded shortening at the start of stimulation to remove the stress that would normally be developed in fixed-end conditions, and used X-ray diffraction to determine the resulting effects on the structure of the thick filaments and the conformation of the myosin motors. The mechano-sensing paradigm was initially proposed based on the results from similar experiments on single fibres from amphibian muscle (Linari et al., 2015), but these X-ray protocols have not been applied previously to mammalian muscle. The present experiments have both strengths and limitations with respect to the previous study. The combination of the narrow X-ray beam and a large, high-resolution X-ray detector used here with the relatively strong diffraction from mouse EDL muscle provided higher signal-to-noise, and improved temporal and spatial resolution. This allowed us to extend the analysis of the narrow and closely spaced interference sub-peaks of those meridional X-ray reflections, and thereby obtain information about motor conformations and the location of the ordered motors in the thick filament. On the other hand, the larger series compliance of the whole muscle preparation means that there is more shortening in the fixed-end periods of force development, which must be taken into account in the interpretation.

In the experiments described above, ramp shortening was applied for 10 ms from the time of the first stimulus, and X-ray data were acquired from 0−3 ms, 5−8 ms, 10−13 ms, 15−18 ms, etc., with the gaps between acquisition windows being required for data transfer from the X-ray detector. The 0−3 ms window corresponds to the mechanical latency, and is also a structural latency period; there were no detectable changes in the X-ray reflections in that window compared with resting conditions. In the 5−8 ms window there was no detectable tension in the ramp shortening protocol, in which the muscle is shortening at its maximum unloaded velocity ($V_0$). Some small but reproducible changes in the X-ray pattern were detected, however, and in particular, the periodicity of the thick filament backbone ($S_{M6}$; Fig. 4D, filled green circle) increased by about 10% of the change between its resting and tetanus plateau value, whereas the axial periodicity of the myosin motors decreased ($S_{M3}$; Fig. 5D, filled green circle). The equatorial intensity ratio also increased ($I_{1,1}/I_{1,0}$; Fig. 2F), which might indicate motors moving away from the filament backbone, although an effect of sarcomere shortening *per se* on this parameter (Reconditi et al., 2014) cannot be excluded.

Larger changes were observed in the 10−13 ms window (magenta). Although this is immediately after the imposed shortening ramp, sarcomere shortening continues as a result of the large series compliance in the mouse EDL muscle, and sarcomeres have shortened by about 12% at the isometric plateau force $T_0$ (Hill et al., 2021). The mean force in the 10−13 ms window after the ramp was only about 3.5% $T_0$ (Table 2), and sarcomeres are expected to be shortening at more than 90% $V_0$ under this load (Brooks & Faulkner, 1988). $S_{M6}$ had increased by about 24% in this period, and $I_{1,1}/I_{1,0}$ by 20%. The decrease in $S_{M3}$ observed in the previous time window was maintained, and $A_{ML1}$ had decreased by 19% (Table 2).

These results suggest some modifications of the mechano-sensing paradigm. The imposition of unloaded shortening at the start of electrical stimulation delays the activation of the thick filament, as in amphibian muscle (Linari et al., 2015), but there is detectable activation at very low load. This is not likely to reflect a fundamental difference between mammalian and amphibian fast-twitch muscles, because the increase in $S_{M6}$ and decrease in $S_{M3}$ were observed in the latter preparation during unloaded shortening starting 5ms after the start of stimulation using a first-generation time-resolved 2D X-ray detector (Brunello et al., 2006). The greater signal-to-noise ratio obtained with whole mouse EDL, combined with a new generation of time-resolved X-ray detectors, has made these small changes in $S_{M6}$ and $S_{M3}$ (considered further below) easier to resolve. Even at zero filament strain, the characteristic OFF or resting structure of the thick filament alters when the thin filament is activated, and in particular, the

thick filament backbone elongates slightly, as signalled by the increase in $S_{M6}$. This change is much smaller than that observed at high force; the major determinant of myosin filament activation is filament stress, matching the number of active motors to the external load as in the original mechano-sensing paradigm (Linari et al., 2015). However, our results suggest the presence of an additional signalling pathway that can produce a low level of activation of the thick filament in the absence of filament strain when the *thin* filament is activated.

The higher resolution of the present measurements also allowed us to interpret the interference fine structure of the M3 reflection in terms of a simple model of the distribution of ordered motors as a periodic array in each half thick filament (Figs 6 and 7). $\Delta z$, which reports the motion of the average centre-of-mass of the ordered motors with respect to its value in resting muscle when the motors are folded back against the filament backbone, increases to about 11 nm at the plateau of an isometric tetanus, in which motors take up an actin-bound conformation roughly perpendicular to the filament axis (Hill et al., 2021; Reconditi et al., 2011). $\Delta z$ was remarkably constant both during the shortening ramp, from 5 to 8 ms (Fig. 7*A*, filled light green circle), and immediately after it, from 10−13 ms (filled magenta circle), when sarcomeres continue to shorten at near-maximum velocity against tendon compliance. $\Delta z$ also retained its resting value in the 5−8 ms window in fixed-end conditions (open light green circle), when force had reached 16% $T_0$. Motors leave the folded helical conformation in all these time windows, as shown by the concomitant decreases in $A_{ML1}$ (Fig. 3*F*) and $A_{M3}$ (Table 2), but become axially disordered, so the observed constancy of HBZ indicates that the motors that remain ordered also remain folded. Although a small number of motors are driving filament sliding at these low loads, no motors with the much larger value of $\Delta z$ corresponding to the perpendicular conformation at the isometric tetanus plateau were detected. This may be understood in terms of the wide range of actin-attached motor conformations during filament sliding, which makes little contribution to the M3 X-ray reflection because their axial mass distribution is broad compared with the 14.5-nm repeat (Huxley et al., 2006; Piazzesi et al., 2007).

The switch to the isometric force-generating conformation of the motors takes place remarkably quickly as force develops and sarcomere shortening slows. In the fixed-end protocol, $\Delta z$ has already increased by about 4 nm in the 10−13 ms window (Fig. 7*A*, open magenta circle), when force is 33% $T_0$ and has essentially reached the isometric plateau value in the 15−18 ms window (open blue circle) when force is only 52% $T_0$. Essentially the same changes occur in the ramp shortening protocol (filled circles) but are delayed by about 5 ms (Table 1). This switch-like change in $\Delta z$ during early

force development suggests a highly cooperative transition between the folded helical motor conformation and the isometric force-generating conformation which is complete in only 10 ms. Such a cooperative transition is consistent with the positive feedback loop implied by the mechano-sensing paradigm. Activation of the thick filaments as signalled by $\Delta z$ is complete at about 50% $T_0$, consistent with the previous inference from the time course of $S_{M6}$ in frog muscle fibres (Linari et al., 2015). The later part of force development takes place with little or no change in $\Delta z$; the axial motion of the centre of mass associated with force generation in the actin-attached motor (Piazzesi et al., 2002) is much smaller than that associated with the loss of the folded conformation (Reconditi et al., 2011).

The interference fine structure of the M3 reflection also gives information about the location of the axially ordered motors in each half myosin filament (Fig. 7*C*). If ordered motors were confined to a small region of each half filament, the shorter diffracting array would produce an axially broader M3 reflection, although this would still be sampled by X-ray interference between the two diffracting arrays in each filament (Brunello et al., 2020; Reconditi, 2006). This possibility was incorporated into the modelling of the axial profile of the M3 reflection in terms of the fitted parameters $n_m$ and $n_d$, the most medial and most distal diffracting motors in the diffracting array (Fig. 7*C* inset). If all the motors in the half filament were axially ordered, $n_m$ would be 1 and $n_d$ would be 49. Within the precision of the measurements, this was the case both in resting muscle (Fig. 7*C*, cyan) and at the isometric tetanus plateau (red). $n_d$ appears to decrease transiently at the start of activation, and this trend was clearer in the low-load shortening protocol, in which the distal five layers of motors become disordered during and immediately after ramp shortening (Fig. 7*C*, filled light green and magenta circles). This suggests that the small number of motors that drive filament sliding under these conditions, the constitutively ON or sentinel motors (Craig & Padrón, 2022; Linari et al., 2015), are located at the filament tips. This would also be plausible from both a structural and a functional perspective; the tip motors have fewer molecular interactions with their neighbour myosins in the folded state, and preferential early activation of the tip motors would pull the remainder of the half filament towards the Z-line of the sarcomere. In terms of the mechano-sensing paradigm, tip motors are in the optimal position to stress and activate the more medial regions of the half filament in the presence of an external load.

The subsequent transient increase in $n_m$, indicating a transient decrease in the order of the most medial motors at 10−13 ms in fixed-end contractions (Fig. 7*C*, open magenta square) and 5 ms later in contractions with low-load shortening (filled blue square) is consistent with

the above picture of sequential zonal activation of the myosin filament. In each case the transient increase in $n_m$ is simultaneous with the switch in $\Delta z$ (Fig. 7*A*), suggesting that the medial (P-zone, Fig. 7*C* inset) motors become transiently disordered during the transition from the folded to the actin-bound force-generating conformation. The $n_m$ transient occurs when fibre force is 33% $T_0$ in the fixed-end and 25% $T_0$ in the low-load shortening protocol, but filament stress must increase from the filament tip to its midpoint, leading to the preferentially earlier activation of P-zone motors.

The transient decrease in $S_{M3}$ (Fig. 5*D*) and in the fitted axial periodicity $d$ in the model of the motor array (Fig. 7*B*) during and immediately after low load shortening was not clearly resolved in previous X-ray experiments on single fibres from amphibian muscle in either fixed-end contractions (Reconditi et al., 2011) or when low-load shortening was imposed soon after the start of stimulation (Linari et al., 2015). However, those studies used a detector with slow readout, and time-series data were reconstructed from a series of snapshots taken using a fast shutter opening at different times during repeated tetani, with each X-ray exposure at a different place along the muscle fibre. As a result, small transient changes in spacing would have been difficult to detect. The transient decrease in $S_{M3}$ is observed at a time when $S_{M6}$, and therefore presumably the axial periodicity of the filament backbone, is *increasing*. A decrease in $S_{M3}$ of about the same size was observed when relaxed demembranated fibres from rabbit psoas muscles (Caremani et al., 2021) or demembranated cardiac trabeculae from rat hearts (Ovejero et al., 2022) were cooled from 35 to 20°C, another protocol that reproduces some of the structural effects of myosin filament activation in the absence of filament stress. Moreover, cooling relaxed demembranated muscle fibres and cardiac trabeculae also leads to the appearance of the 'star' peak on the low angle side of the M3 reflection, suggesting that the two phenomena are linked, perhaps through X-ray interference between the *ca* 14.34-nm axial periodicity responsible for the M3 reflection and the slightly longer *ca* 45.5-nm periodicity that contributes to the M1 and M2 reflections. Another possibility is that the tips of the filaments have a slightly longer axial periodicity than the more central regions and that the loss of a few layers of ordered tip motors during unloaded shortening (Fig. 7*C*) is responsible for the decrease in the observed $S_{M3}$ (Brunello et al., 2020; Ovejero et al., 2022).

In summary, the above results demonstrate that filament stress is the major but not the only mechanism of myosin filament activation in mammalian fast-twitch muscle, and reveal that activation has distinct dynamics in different zones of the myosin filament. The improved understanding of the control of muscle strength via the fraction of active motors produced by these findings may underpin the development of new approaches and assays for potential therapeutics for muscle weakness based on targeting the mechanisms that control that fraction.

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

## Additional information

### Data availability statement

The data generated and analysed in this study are available from the corresponding author on request.

### Competing interests

The authors declare no conflict of interest.

### Author contributions

Conceptualization: C.H., E.B., L.F., M.I. Formal analysis: C.H., E.B., L.F., M.I. Investigation: C.H., E.B., L.F., M.I. and J.G.O.

Visualization: C.H., E.B., L.F., M.I. Methodology: C.H., E.B., L.F., M.I. Writing – original draft: C.H., E.B., L.F., M.I. Writing – review and editing: C.H., E.B., L.F., M.I. and J.G.O. Project administration: C.H., E.B., L.F., M.I. Supervision: E.B., L.F., M.I. Funding acquisition: E.B., L.F., M.I. All authors approved the final version of the manuscript and agree to be accountable for all aspects of the work in ensuring that questions related to the accuracy or integrity of any part of the work are appropriately investigated and resolved. All persons designated as authors qualify for authorship, and all those who qualify for authorship are listed.

## Funding

This work was funded by the Medical Research Council MR/R01700X/1 and Diamond Light Source (SM21316-1). E.B. and J.G.O. were supported by a British Heart Foundation Intermediate Basic Science Research Fellowship awarded to E.B. (FS/17/3/32604). L.F. was funded by a Sir Henry Dale Fellowship awarded by the Wellcome Trust and the Royal Society (210464/Z/18/Z).

## Acknowledgements

We thank I22 staff, Olga Shebanova, Tim Snow and Nick Terrill (Diamond Light Source) and MRC Harwell for support during the beamtime; Diamond Light Source for the provision of synchrotron beamtime; Marty Rajaratnam (King's College London) for mechanical engineering support and Thomas Kavanagh (King's College London) for helping with 3D-printing.

## Keywords

muscle regulation, myosin, skeletal muscle

## Supporting information

Additional supporting information can be found online in the Supporting Information section at the end of the HTML view of the article. Supporting information files available:

**Peer Review History**
**Statistical Summary Document**

