## [Peer Review History · The Journal of Physiology]

Activation of the myosin motors in fast-twitch muscle of the mouse is controlled by mechano-sensing in the myosin filaments

Cameron Hill, Elisabetta Brunello, Luca Fusi, Jesús Garcia Ovejero, and Malcolm Irving

DOI: 10.1113/JP283048

Corresponding author(s): Cameron Hill (cameron.hill@kcl.ac.uk)

Review Timeline:

Submission Date:	16-Mar-2022
Editorial Decision:	19-Apr-2022
Revision Received:	29-Jun-2022
Editorial Decision:	15-Jul-2022
Revision Received:	22-Jul-2022
Accepted:	27-Jul-2022

Senior Editor: Michael Hogan

Reviewing Editor: Wolfgang Linke

Transaction Report:

Dear Dr Hill,

Re: JP-RP-2022-283048 "Activation of the myosin motors in fast-twitch muscle of the mouse is controlled by mechano-sensing in the myosin filaments" by Cameron Hill, Elisabetta Brunello, Luca Fusi, Jesús Garcia Ovejero, and Malcolm Irving

Thank you for submitting your manuscript to The Journal of Physiology. It has been assessed by a Reviewing Editor and by 2 expert Referees and I am pleased to tell you that it is considered to be acceptable for publication following satisfactory revision.

The reports are copied at the end of this email. Please address all of the points and incorporate all requested revisions, or explain in your Response to Referees why a change has not been made.

NEW POLICY: In order to improve the transparency of its peer review process The Journal of Physiology publishes online as supporting information the peer review history of all articles accepted for publication. Readers will have access to decision letters, including all Editors' comments and referee reports, for each version of the manuscript and any author responses to peer review comments. Referees can decide whether or not they wish to be named on the peer review history document.

Authors are asked to use The Journal's premium BioRender (<https://biorender.com/>) account to create/redraw their Abstract Figures. Information on how to access The Journal's premium BioRender account is here:

<https://physoc.onlinelibrary.wiley.com/journal/14697793/biorender-access> and authors are expected to use this service. This will enable Authors to download high-resolution versions of their figures. The link provided should only be used for the purposes of this submission. Authors will be charged for figures created on this premium BioRender account if they are not related to this manuscript submission.

I hope you will find the comments helpful and have no difficulty returning your revisions within 4 weeks.

Your revised manuscript should be submitted online using the links in Author Tasks Link Not Available.

Any image files uploaded with the previous version are retained on the system. Please ensure you replace or remove all files that have been revised.

REVISION CHECKLIST:

- Article file, including any tables and figure legends, must be in an editable format (eg Word)
- Abstract figure file (see above)
- Statistical Summary Document
- Upload each figure as a separate high quality file
- Upload a full Response to Referees, including a response to any Senior and Reviewing Editor Comments;
- Upload a copy of the manuscript with the changes highlighted.

- A potential 'Cover Art' file for consideration as the Issue's cover image;
- Appropriate Supporting Information (Video, audio or data set https://jp.msubmit.net/cgi-bin/main.plex?form_type=display_requirements#supp).

To create your 'Response to Referees' copy all the reports, including any comments from the Senior and Reviewing Editors, into a Word, or similar, file and respond to each point in colour or CAPITALS and upload this when you submit your revision.

I look forward to receiving your revised submission.

If you have any queries please reply to this email and staff will be happy to assist.

Yours sincerely,

Michael C. Hogan
Senior Editor
The Journal of Physiology
<https://jp.msubmit.net>
<http://jp.physoc.org>
The Physiological Society
Hodgkin Huxley House
30 Farringdon Lane
London, EC1R 3AW
UK
<http://www.physoc.org>
<http://journals.physoc.org>

REQUIRED ITEMS:

- Author photo and profile. First (or joint first) authors are asked to provide a short biography (no more than 100 words for one author or 150 words in total for joint first authors) and a portrait photograph. These should be uploaded and clearly labelled with the revised version of the manuscript. See Information for Authors for further details.
- You must start the Methods section with a paragraph headed Ethical Approval. A detailed explanation of journal policy and regulations on animal experimentation is given in Principles and standards for reporting animal experiments in The Journal of Physiology and Experimental Physiology by David Grundy J Physiol, 593: 2547-2549. doi:10.1113/JP270818.). A checklist outlining these requirements and detailing the information that must be provided in the paper can be found at: <https://physoc.onlinelibrary.wiley.com/hub/animal-experiments>. Authors should confirm in their Methods section that their experiments were carried out according to the guidelines laid down by their institution's animal welfare committee, and conform to the principles and regulations as described in the Editorial by Grundy (2015). The Methods section must contain details of the anaesthetic regime: anaesthetic used, dose and route of administration and method of killing the experimental animals.
- Please upload separate high-quality figure files via the submission form.
- Please ensure that any tables are in Word format and are, wherever possible, embedded in the article file itself.
- Please ensure that the Article File you upload is a Word file.
- A Statistical Summary Document, summarising the statistics presented in the manuscript, is required upon revision. It must be on the Journal's template, which can be downloaded from the link in the Statistical Summary Document section here: https://jp.msubmit.net/cgi-bin/main.plex?form_type=display_requirements#statistics
- Papers must comply with the Statistics Policy https://jp.msubmit.net/cgi-bin/main.plex?form_type=display_requirements#statistics

In summary:

- If $n \leq 30$, all data points must be plotted in the figure in a way that reveals their range and distribution. A bar graph with data points overlaid, a box and whisker plot or a violin plot (preferably with data points included) are acceptable formats.
- If $n > 30$, then the entire raw dataset must be made available either as supporting information, or hosted on a not-for-profit repository e.g. FigShare, with access details provided in the manuscript.
- 'n' clearly defined (e.g. x cells from y slices in z animals) in the Methods. Authors should be mindful of pseudoreplication.
- All relevant 'n' values must be clearly stated in the main text, figures and tables, and the Statistical Summary Document (required upon revision)
- The most appropriate summary statistic (e.g. mean or median and standard deviation) must be used. Standard Error of the Mean (SEM) alone is not permitted.
- Exact p values must be stated. Authors must not use 'greater than' or 'less than'. Exact p values must be stated to three significant figures even when 'no statistical significance' is claimed.
- Statistics Summary Document completed appropriately upon revision.

- Please include an Abstract Figure. The Abstract Figure is a piece of artwork designed to give readers an immediate understanding of the research and should summarise the main conclusions. If possible, the image should be easily 'readable' from left to right or top to bottom. It should show the physiological relevance of the manuscript so readers can assess the importance and content of its findings. Abstract Figures should not merely recapitulate other figures in the manuscript. Please try to keep the diagram as simple as possible and without superfluous information that may distract from the main conclusion(s). Abstract Figures must be provided by authors no later than the revised manuscript stage and should be uploaded as a separate file during online submission labelled as File Type 'Abstract Figure'. Please ensure that you include the figure legend in the main article file. All Abstract Figures should be created using BioRender. Authors should use The Journal's premium BioRender account to export high-resolution images. Details on how to use and access the premium account are included as part of this email.

EDITOR COMMENTS

Reviewing Editor:

We are pleased that your manuscript is acceptable for publication if you properly address all comments of the reviewers. Please take note of the request to include examples of diffraction patterns as this illustrates the quality of your experiments and helps the readers to better understand the methodology.

REFEREE COMMENTS

Referee #1:

The manuscript by Hill et al studies the activation of myosin motors in EDL muscle of the mouse. The study is follow-up of previous work on frog muscle, published by the same group in 2015. The data presented here shows that also in mammalian muscle myosin is predominantly activated via filament stress. Interestingly, the current data also suggests that filament 'zonal' dynamics play a role in activation (thus, independent of filament stress). The approach is state-of-the-art and combines x-ray diffraction with muscle mechanics protocols to manipulate filament stress during muscle activation. The findings are important and improve our understanding of muscle structure and function. I have a few comments the authors should address.

Lines 77-80: the authors discuss here maximal tetanic activation? Because, during twitch contractions, calcium transients are of course small and the resulting force as well. This should be rephrased as readers not so familiar with skeletal muscle activation might be confused by these statements.

Lines 176-177: for the unloaded shortening protocol, the muscle is first stretched by 5% from L_0 , held for 300ms, and then activated and allowed to shorten. The 5% stretch will increase passive tension, I assume? If so, the structures responsible for passive tension (titin?) might pull on the myosin filament and affect its structure. It is not clear to me whether such passive tension develops during the protocol. The authors should discuss this in the manuscript. And if this passive tension develops they should discuss its potential effect on the outcome of the data.

Lines 183-186: To minimize noise, muscles were activated many times, up to 16 consecutive contractions. They were all maximal tetani. Such a protocol might induce movement of the sutures relative to the tendons and as a result sarcomere length might become shorter. Do the authors think this occurs and if so, does this affect the outcomes?

Figures, general: the figures are easy to understand (as far as that is possible when it comes to x-ray data...), but I was surprised that no examples of diffraction patterns are shown. In my opinion, typical examples that illustrate the reflections would benefit the manuscript and help the reader understand the data. Furthermore, a schematic of a sarcomere that illustrates which reflections 'reflect' which structures in the myosin filament would also increase the understanding of the complex data-sets.

Lines 363: the S_{ml1} values, are they significantly different (43.0 vs 42.8)?

Lines 718-720: the authors indicate that sarcomere shortening might affect the $I_{1,1}/I_{1,0}$ data. Can the authors speculate on how this might be affected by sarcomere length?

Lines 776-778: how can force further develop if the proportions of folded and force generating motors are constant?

Lines: 831-833: How can these findings add to the development of therapeutics? The authors could be a bit more specific here.

Referee #2:

The manuscript of Hill et al aims to extend information on the mechanism of activation of the thick filaments in fast-twitch

mammalian skeletal muscles contracting in near-physiological conditions by applying unloaded shortening during the first 10 ms after the start of electrical stimulation and comparing the structural changes in the low angle X-ray diffraction pattern obtained at high spatial and temporal resolution with those during fixed-end recordings. The experiments are performed with care and the presentation and discussion of the data, modelling and interpretation is very clear. A comprehensive description is provided of the equatorial 1,0 and 1,1 reflections, the ML1 and AL1 reflections, and the meridional M6 and M3 reflections and the changes thereof during fixed-end and unloaded shortening during tetanic force development in fast murine muscle. The main novel findings of this study are that even at zero filament strain there is a very early structural change in the myosin head configuration and a slight elongation of the thick filament backbone. The modelling results indicate that the early changes in myosin head configuration occur in the more distal crowns of myosin heads. Below please find some questions and minor comments:

Major

1. The half bare zone (hbz) width increases from approximately 80 at rest to 91 nm at maximum tetanic force development. Can this 11 nm (~14%) increase be fully attributed to the change in the centre of mass of the first group of myosin heads next to the bare zone?
2. Could the increase in the cytosolic calcium concentration play a role in the (zonal) activation of the thick filament? To discuss this issue the position of the terminal cisternae at the A-I junction where calcium is released could be taken into consideration.
3. Is the mass of Myosin Binding protein C (in the C-zone (Fig. 6)) negligible in the model calculations?

Minor comments

- I. 402 Since the M6 reflection consists of different peaks, which one defines SM6?
- I. 663 Were some of the experiments performed in a paired fashion, i.e. fixed-end and unloaded shortening in the same muscle? Please explain "a".

Fig. 6D. In the PDF copy of the diagram the layers of myosin heads at the tip are not equidistant. Please check the original.

END OF COMMENTS

Confidential Review

16-Mar-2022

EDITOR COMMENTS

Reviewing Editor:

We are pleased that your manuscript is acceptable for publication if you properly address all comments of the reviewers. Please take note of the request to include examples of diffraction patterns as this illustrates the quality of your experiments and helps the readers to better understand the methodology.

REFEREE COMMENTS

Referee #1:

The manuscript by Hill et al studies the activation of myosin motors in EDL muscle of the mouse. The study is follow-up of previous work on frog muscle, published by the same group in 2015. The data presented here shows that also in mammalian muscle myosin is predominantly activated via filament stress. Interestingly, the current data also suggests that filament 'zonal' dynamics play a role in activation (thus, independent of filament stress). The approach is state-of-the-art and combines x-ray diffraction with muscle mechanics protocols to manipulate filament stress during muscle activation. The findings are important and improve our understanding of muscle structure and function. I have a few comments the authors should address.

Response – Thank you for taking the time to review our manuscript. We are pleased you find the results of importance and have worked to address your concerns.

Lines 77-80: the authors discuss here maximal tetanic activation? Because, during twitch contractions, calcium transients are of course small and the resulting force as well. This should be rephrased as readers not so familiar with skeletal muscle activation might be confused by these statements.

Response - Calcium transients recorded in the preparation and conditions of the present experiments reach concentrations much larger than the dissociation constant of troponin, even in a twitch. Moreover, although the calcium transient is brief in a twitch, the ON rate is sufficiently fast that the calcium regulatory sites of troponin become fully occupied (Baylor & Hollingsworth, 2003).

Lines 176-177: for the unloaded shortening protocol, the muscle is first stretched by 5% from L_0 , held for 300ms, and then activated and allowed to shorten. The 5% stretch will increase passive tension, I assume? If so, the structures responsible for passive tension (titin?) might pull on the myosin filament and affect its structure. It is not clear to me whether such passive tension develops during the protocol. The authors should discuss this in the manuscript. And if this passive tension develops they should discuss its potential effect on the outcome of the data.

Response - The passive tension is very small at these sarcomere lengths, but the values are now added to Table 2.

Lines 183-186: To minimize noise, muscles were activated many times, up to 16 consecutive contractions. They were all maximal tetani. Such a protocol might induce movement of the sutures relative to the tendons and as a result sarcomere length might become shorter. Do the authors think this occurs and if so, does this affect the outcomes?

Response - With the tetanus duration and protocols used here any slippage of the sutures occurs in the first series of tetani used to set the initial muscle length, before the series of tetani used for the X-ray experiments and is detectable in the mechanical records. Such slippage was not observed in the later series of tetani used for the X-ray experiments, during which the mechanical response was reproducible, and plateau tension declined by less than 10% during the series.

Figures, general: the figures are easy to understand (as far as that is possible when it comes to x-ray data...), but I was surprised that no examples of diffraction patterns are shown. In my opinion, typical examples that illustrate the reflections would benefit the manuscript and help the reader understand the data. Furthermore, a schematic of a sarcomere that illustrates which reflections 'reflect' which structures in the myosin filament would also increase the understanding of the complex data-sets.

Response - A new Figure (Fig. 1) containing an example diffraction pattern and a schematic has been added.

Lines 363: the Sml1 values, are they significantly different (43.0 vs 42.8)?

Response - They are not significantly different, now noted in the text on lines 325-326.

Lines 718-720: the authors indicate that sarcomere shortening might affect the I1,1/I1,0 data. Can the authors speculate on how this might be affected by sarcomere length?

Response - This was an empirical statement, with a reference to a paper (Reconditi et al 2014) that shows the effect. The most likely mechanism is the greater length of thin filament in the filament overlap zone of each sarcomere, but the point being made here is independent of the mechanism, so we prefer not to speculate.

Lines 776-778: how can force further develop if the proportions of folded and force generating motors are constant?

Response - This statement was an oversimplification, and we have revised it (lines 511-514) to take into account that there may be more than two structural and functional states of the myosin motors. However, the axial movement of the centre of mass of the motors associated with the force generating transition in actin-attached motors during the last part of force development is very small (ca 1nm; Piazzesi et al., 2002, 2007) compared with the transition between the folded and actin-attached states (ca 11nm; Reconditi et al 2011).

Lines: 831-833: How can these findings add to the development of therapeutics? The authors could be a bit more specific here.

Response - We expanded the argument on lines 567-570 to make the point that the understanding that muscle strength is largely limited by the fraction of ON motors allows a novel therapeutic approach based on targeting the control of that fraction.

Referee #2:

The manuscript of Hill et al aims to extend information on the mechanism of activation of the thick filaments in fast-twitch mammalian skeletal muscles contracting in near-physiological conditions by applying unloaded shortening during the first 10 ms after the start of electrical stimulation and comparing the structural changes in the low angle X-ray diffraction pattern obtained at high spatial and temporal resolution with those during fixed-end recordings. The experiments are performed with care and the presentation and discussion of the data, modelling and interpretation is very clear. A comprehensive description is provided of the equatorial 1,0 and 1,1 reflections, the ML1 and AL1 reflections, and the meridional M6 and M3 reflections and the changes thereof during fixed-end and unloaded shortening during tetanic force development in fast murine muscle. The main novel findings of this study are that even at zero filament strain there is a very early structural change in the myosin head configuration and a slight elongation of the thick filament backbone. The modelling results indicate that the early changes in myosin head configuration occur in the more distal crowns of myosin heads. Below please find some questions and minor comments:

Response – Thank you for your time to review our manuscript. We hope to have satisfactorily addressed your below comments.

Major

1. The half bare zone (hbz) width increases from approximately 80 at rest to 91 nm at maximum tetanic force development. Can this 11 nm (~14%) increase be fully attributed to the change in the centre of mass of the first group of myosin heads next to the bare zone?

Response - The parameter 'hbz' in the model represents the average centre of mass of all the ordered motors. The submitted text may have been confusing in this respect and has been amended to be more explicit. More generally, the reviewer's comment made us realise that it would be clearer to express this model parameter as 'delta z' (Δz) to avoid this possible confusion. The text (lines 264-266) and Fig. 7 have been amended accordingly.

2. Could the increase in the cytosolic calcium concentration play a role in the (zonal) activation of the thick filament? To discuss this issue the position of the terminal cisternae at the A-I junction where calcium is released could be taken into consideration.

Response - This has been modelled by Baylor & Hollingsworth (JGP 2007). Calcium release at the AI junction rather than the Z-line does make thin filament activation more uniform and synchronous, but the difference in the half activation times for troponin across the thin filament is only about 1ms at sarcomere length 2.4 μ m, 16degC (see their Fig 13), and would be only a fraction of 1ms in our experiments at 27degC, much faster than the observed zonal effects.

3. Is the mass of Myosin Binding protein C (in the C-zone (Fig. 6)) negligible in the model calculations?

Response - We did not include MyBP-C in the model, because its conformation is unknown. Its contribution is likely to be small though, even in the C zone, because of its lower stoichiometry and the immobility of its C-terminal domains.

Minor comments

I. 402 Since the M6 reflection consists of different peaks, which one defines SM6?

Response - SM6 is the weighted average of all the interference sub-peaks (Methods II. 206-210)

I. 663 Were some of the experiments performed in a paired fashion, i.e. fixed-end and unloaded shortening in the same muscle? Please explain "a".

Response - The fixed-end and unloaded shortening protocols were not performed in the same muscle. The 'a' in the first row of Table 1 was a mistake, and the meaning of the paired comparisons in the Tables, as a comparison with the corresponding parameter for force, has been made explicit in the Table legends.

Fig. 6D. In the PDF copy of the diagram the layers of myosin heads at the tip are not equidistant. Please check the original.

Response - This was intentional; the third layer of motors from the tip is always missing in the mammalian thick filament.

END OF COMMENTS

Dear Dr Hill,

Re: JP-RP-2022-283048R1 "Activation of the myosin motors in fast-twitch muscle of the mouse is controlled by mechanosensing in the myosin filaments" by Cameron Hill, Elisabetta Brunello, Luca Fusi, Jesús Garcia Ovejero, and Malcolm Irving

Thank you for submitting your manuscript to The Journal of Physiology. It has been assessed by a Reviewing Editor and by 2 expert Referees and I am pleased to tell you that it is considered to be acceptable for publication following satisfactory revision.

The reports are copied at the end of this email. Please address all of the points and incorporate all requested revisions, or explain in your Response to Referees why a change has not been made.

NEW POLICY: In order to improve the transparency of its peer review process The Journal of Physiology publishes online as supporting information the peer review history of all articles accepted for publication. Readers will have access to decision letters, including all Editors' comments and referee reports, for each version of the manuscript and any author responses to peer review comments. Referees can decide whether or not they wish to be named on the peer review history document.

Authors are asked to use The Journal's premium BioRender (<https://biorender.com/>) account to create/redraw their Abstract Figures. Information on how to access The Journal's premium BioRender account is here: <https://physoc.onlinelibrary.wiley.com/journal/14697793/biorender-access> and authors are expected to use this service. This will enable Authors to download high-resolution versions of their figures. The link provided should only be used for the purposes of this submission. Authors will be charged for figures created on this premium BioRender account if they are not related to this manuscript submission.

I hope you will find the comments helpful and have no difficulty returning your revisions within 4 weeks.

Your revised manuscript should be submitted online using the links in Author Tasks Link Not Available.

Any image files uploaded with the previous version are retained on the system. Please ensure you replace or remove all files that have been revised.

REVISION CHECKLIST:

- Article file, including any tables and figure legends, must be in an editable format (eg Word)
- Abstract figure file (see above)
- Statistical Summary Document
- Upload each figure as a separate high quality file
- Upload a full Response to Referees, including a response to any Senior and Reviewing Editor Comments;
- Upload a copy of the manuscript with the changes highlighted.

- A potential 'Cover Art' file for consideration as the Issue's cover image;
- Appropriate Supporting Information (Video, audio or data set https://jp.msubmit.net/cgi-bin/main.plex?form_type=display_requirements#supp).

To create your 'Response to Referees' copy all the reports, including any comments from the Senior and Reviewing Editors, into a Word, or similar, file and respond to each point in colour or CAPITALS and upload this when you submit your revision.

I look forward to receiving your revised submission.

If you have any queries please reply to this email and staff will be happy to assist.

Yours sincerely,

Michael C. Hogan
Senior Editor
The Journal of Physiology
<https://jp.msubmit.net>
<http://jp.physoc.org>
The Physiological Society
Hodgkin Huxley House
30 Farringdon Lane
London, EC1R 3AW
UK
<http://www.physoc.org>
<http://journals.physoc.org>

REQUIRED ITEMS:

- Author profile. First (or joint first) authors are asked to provide a short biography (no more than 100 words for one author or 150 words in total for joint first authors). This should be uploaded and clearly labelled with the revised version of the manuscript. See Information for Authors for further details.

- Abstract figure: The Journal of Physiology funds authors of provisionally accepted papers to use the premium BioRender site to create high resolution schematic figures. Follow this link and enter your details and the manuscript number to create and download figures. Upload these as the figure files for your revised submission. If you choose not to take up this offer we require figures to be of similar quality and resolution. If you are opting out of this service to authors, state this in the Comments section on the Detailed Information page of the submission form. The link provided should only be used for the purposes of this submission. Authors will be charged for figures created on this premium BioRender account if they are not related to this manuscript submission.

- You must start the Methods section with a paragraph headed Ethical Approval. A detailed explanation of journal policy and regulations on animal experimentation is given in Principles and standards for reporting animal experiments in The Journal of Physiology and Experimental Physiology by David Grundy J Physiol, 593: 2547-2549. doi:10.1113/JP270818). A checklist outlining these requirements and detailing the information that must be provided in the paper can be found at: <https://physoc.onlinelibrary.wiley.com/hub/animal-experiments>. Authors should confirm in their Methods section that their experiments were carried out according to the guidelines laid down by their institution's animal welfare committee, and conform to the principles and regulations as described in the Editorial by Grundy (2015). The Methods section must contain details of the anaesthetic regime: anaesthetic used, dose and route of administration and method of killing the experimental animals.

- Papers must comply with the Statistics Policy: https://jp.msubmit.net/cgi-bin/main.plex?form_type=display_requirements#statistics.

In summary:

- If $n \leq 30$, all data points must be plotted in the figure in a way that reveals their range and distribution. A bar graph with data points overlaid, a box and whisker plot or a violin plot (preferably with data points included) are acceptable formats.

- If $n > 30$, then the entire raw dataset must be made available either as supporting information, or hosted on a not-for-profit repository e.g. FigShare, with access details provided in the manuscript.

- 'n' clearly defined (e.g. x cells from y slices in z animals) in the Methods. Authors should be mindful of pseudoreplication.

- All relevant 'n' values must be clearly stated in the main text, figures and tables, and the Statistical Summary Document (required upon revision).

- The most appropriate summary statistic (e.g. mean or median and standard deviation) must be used. Standard Error of the Mean (SEM) alone is not permitted.

- Exact p values must be stated. Authors must not use 'greater than' or 'less than'. Exact p values must be stated to three significant figures even when 'no statistical significance' is claimed.

EDITOR COMMENTS

Reviewing Editor:

Your manuscript is acceptable for publication if you address the minor points raised by reviewer 2.

REFEREE COMMENTS

Referee #1:

The authors addressed my remarks and I have no further comments.

Referee #2:

I have one suggestion and two questions regarding the response in the rebuttal:

Major point 2

Could the increase in the cytosolic calcium concentration play a role in the (zonal) activation of the thick filament? To discuss this issue the position of the terminal cisternae at the A-I junction where calcium is released could be taken into consideration.

Response - This has been modelled by Baylor & Hollingsworth (JGP 2007). Calcium release at the AI junction rather than the Z-line does make thin filament activation more uniform and synchronous, but the difference in the half activation times for troponin across the thin filament is only about 1ms at sarcomere length 2.4um, 16degC (see their Fig 13), and would be only a fraction of 1ms in our experiments at 27degC, much faster than the observed zonal effects.

>>>Suggestion: The authors could consider this issue explicitly in the Discussion of their manuscript by including references to Cannell & Allen (Biophysical Journal 1984, p. 923 (10 ms)) and Previs et al. (Sci Adv 2015;1:e1400205).

Minor comment

Fig. 6D. In the PDF copy of the diagram the layers of myosin heads at the tip are not equidistant. Please check the original.

Response - This was intentional; the third layer of motors from the tip is always missing in the mammalian thick filament.

>>>Questions: Is this feature incorporated in the version of the Reconditi (2006) model used? If not, could this impact the conclusions, because the tip of the myosin filament is where the sentinel motors supposedly are located?

END OF COMMENTS

1st Confidential Review

29-Jun-2022

Major point 2

Could the increase in the cytosolic calcium concentration play a role in the (zonal) activation of the thick filament? To discuss this issue the position of the terminal cisternae at the A-I junction where calcium is released could be taken into consideration.

Response - This has been modelled by Baylor & Hollingsworth (JGP 2007). Calcium release at the AI junction rather than the Z-line does make thin filament activation more uniform and synchronous, but the difference in the half activation times for troponin across the thin filament is only about 1ms at sarcomere length 2.4um, 16degC (see their Fig 13), and would be only a fraction of 1ms in our experiments at 27degC, much faster than the observed zonal effects.

>>>Suggestion: The authors could consider this issue explicitly in the Discussion of their manuscript by including references to Cannell & Allen (Biophysical Journal 1984, p. 923 (10 ms)) and Previs et al. (Sci Adv 2015;1:e1400205).

>>>Response- This issue cannot be relevant to the results in our paper, in which the effective time resolution is 5ms. Baylor and Hollingsworth (see reference above) showed that calcium gradients in the conditions of our experiments would persist for a fraction of 1ms, an order of magnitude less than the time resolution. (Note that the two references mentioned by the reviewer relate to other species and/or muscle types.) Reference to this (in our view 'non-') issue in the Discussion of our paper would distract from the interpretation of our results and the associated conclusions.

Minor comment

Fig. 6D. In the PDF copy of the diagram the layers of myosin heads at the tip are not equidistant. Please check the original.

Response - This was intentional; the third layer of motors from the tip is always missing in the mammalian thick filament.

>>>Questions: Is this feature incorporated in the version of the Reconditi (2006) model used? If not, could this impact the conclusions, because the tip of the myosin filament is where the sentinel motors supposedly are located?

>>> We did not explicitly model the missing layer of heads in the modelling of the axial profile of the M3 reflection, and in that respect we follow all previous published models that used this type of formalism, to our knowledge. Since there are 49 layers of heads in each half filament, the effect of one missing layer near the tip would be very small. The focus of the paper is not on the absolute arrangement of the layers of heads, but on the changes in the ordered layers of heads on stimulation and during unloaded shortening. These changes are much larger than those corresponding to the loss of one layer of heads (Fig.7).

Dear Dr Hill,

Re: JP-RP-2022-283048R2 "Activation of the myosin motors in fast-twitch muscle of the mouse is controlled by mechanosensing in the myosin filaments" by Cameron Hill, Elisabetta Brunello, Luca Fusi, Jesús Garcia Ovejero, and Malcolm Irving

I am pleased to tell you that your paper has been accepted for publication in The Journal of Physiology.

NEW POLICY: In order to improve the transparency of its peer review process The Journal of Physiology publishes online as supporting information the peer review history of all articles accepted for publication. Readers will have access to decision letters, including all Editors' comments and referee reports, for each version of the manuscript and any author responses to peer review comments. Referees can decide whether or not they wish to be named on the peer review history document.

The last Word version of the paper submitted will be used by the Production Editors to prepare your proof. When this is ready you will receive an email containing a link to Wiley's Online Proofing System. The proof should be checked and corrected as quickly as possible.

Authors should note that it is too late at this point to offer corrections prior to proofing. The accepted version will be published online, ahead of the copy edited and typeset version being made available. Major corrections at proof stage, such as changes to figures, will be referred to the Reviewing Editor for approval before they can be incorporated. Only minor changes, such as to style and consistency, should be made a proof stage. Changes that need to be made after proof stage will usually require a formal correction notice.

All queries at proof stage should be sent to TJP@wiley.com.

Are you on Twitter? Once your paper is online, why not share your achievement with your followers. Please tag The Journal (@jphysiol) in any tweets and we will share your accepted paper with our 23,000+ followers!

Yours sincerely,

Michael C. Hogan
Senior Editor
The Journal of Physiology
<https://jp.msubmit.net>
<http://jp.physoc.org>
The Physiological Society
Hodgkin Huxley House
30 Farringdon Lane
London, EC1R 3AW
UK
<http://www.physoc.org>
<http://journals.physoc.org>

P.S. - You can help your research get the attention it deserves! Check out Wiley's free Promotion Guide for best-practice recommendations for promoting your work at www.wileyauthors.com/eeo/guide. And learn more about Wiley Editing Services which offers professional video, design, and writing services to create shareable video abstracts, infographics, conference posters, lay summaries, and research news stories for your research at www.wileyauthors.com/eeo/promotion.

*** IMPORTANT NOTICE ABOUT OPEN ACCESS ***

To assist authors whose funding agencies mandate public access to published research findings sooner than 12 months after publication The Journal of Physiology allows authors to pay an open access (OA) fee to have their papers made freely available immediately on publication.

You will receive an email from Wiley with details on how to register or log-in to Wiley Authors Services where you will be able to place an OnlineOpen order.

You can check if your funder or institution has a Wiley Open Access Account here: <https://authorservices.wiley.com/author-resources/Journal-Authors/licensing-and-open-access/open-access/author-compliance-tool.html>.

Your article will be made Open Access upon publication, or as soon as payment is received.

If you wish to put your paper on an OA website such as PMC or UKPMC or your institutional repository within 12 months of publication you must pay the open access fee, which covers the cost of publication.

OnlineOpen articles are deposited in PubMed Central (PMC) and PMC mirror sites. Authors of OnlineOpen articles are permitted to post the final, published PDF of their article on a website, institutional repository, or other free public server, immediately on publication.

Note to NIH-funded authors: The Journal of Physiology is published on PMC 12 months after publication, NIH-funded authors DO NOT NEED to pay to publish and DO NOT NEED to post their accepted papers on PMC.

EDITOR COMMENTS

The revisions are responsive and the manuscript can now be accepted.

2nd Confidential Review

22-Jul-2022